# Interleukin-18 in Health and Disease

**DOI:** 10.3390/ijms20030649

**Published:** 2019-02-02

**Authors:** Koubun Yasuda, Kenji Nakanishi, Hiroko Tsutsui

**Affiliations:** 1Department of Immunology, Hyogo College of Medicine, 1-1 Mukogawa-cho, Nishinomiya, Hyogo 663-8501, Japan; koubun@hyo-med.ac.jp; 2Department of Surgery, Hyogo College of Medicine, 1-1 Mukogawa-cho, Nishinomiya, Hyogo 663-8501, Japan; tsutsuihiroko3@gmail.com

**Keywords:** IFN-γ, Inflammation, host defense, disease, innate-type allergy, therapeutic target

## Abstract

Interleukin (IL)-18 was originally discovered as a factor that enhanced IFN-γ production from anti-CD3-stimulated Th1 cells, especially in the presence of IL-12. Upon stimulation with Ag plus IL-12, naïve T cells develop into IL-18 receptor (IL-18R) expressing Th1 cells, which increase IFN-γ production in response to IL-18 stimulation. Therefore, IL-12 is a commitment factor that induces the development of Th1 cells. In contrast, IL-18 is a proinflammatory cytokine that facilitates type 1 responses. However, IL-18 without IL-12 but with IL-2, stimulates NK cells, CD4^+^ NKT cells, and established Th1 cells, to produce IL-3, IL-9, and IL-13. Furthermore, together with IL-3, IL-18 stimulates mast cells and basophils to produce IL-4, IL-13, and chemical mediators such as histamine. Therefore, IL-18 is a cytokine that stimulates various cell types and has pleiotropic functions. IL-18 is a member of the IL-1 family of cytokines. IL-18 demonstrates a unique function by binding to a specific receptor expressed on various types of cells. In this review article, we will focus on the unique features of IL-18 in health and disease in experimental animals and humans.

## 1. Introduction

Th1 cells produce interferon (IFN)-γ upon stimulation with antigen (Ag) plus antigen presenting cells or anti-CD3 antibody in vitro and in vivo. Lipopolysaccharide (LPS)-stimulation of anti-CD3-stimulated Th1 cells does not induce the production of IFN-γ in vitro. However, the injection of LPS into *Propionibacterium acnes*-primed mice or *Bacillus Calmette–Guerin* (BCG)-infected mice, but not naïve mice, strongly induced IFN-γ production in vivo [1,2]. Furthermore, to our surprise, the addition of sera derived from *P. acnes*-primed and LPS-challenged mice strongly enhanced IFN-γ production by anti-CD3-stimulated Th1 cells in vitro, suggesting the presence of IFN-γ inducing factor(s) in the sera.

Because IL-12 is produced by LPS-stimulated macrophages and dendritic cells (DC), IL-12 from LPS-stimulated macrophages or DC in *P. acnes*-primed mice were initially thought to induce anti-CD3-stimulated Th1 cells to produce IFN-γ in vivo and in vitro. Indeed, the sera contained high levels of IL-12. However, only the addition of sera from *P. acnes*-primed and LPS-challenged mice, but not the addition of excess doses of IL-12, enhanced the production of IFN-γ from anti-CD3-stimulated Th1 cells, strongly suggesting the presence of IFN-γ inducing factors in the sera from *P. acnes*-primed and LPS-challenged mice.

Physicochemical studies and amino acid sequence analysis revealed that IFN-γ inducing factor (IGIF) is different from IL-12. The molecular cloning of IGIF was performed by Okamura in collaboration with Hayashibara Biochemical Laboratories. Soon after human IGIF was cloned [3], we and others found various functions of IGIF, including the induction of IL-2 production, IL-2 receptor (IL-2R) and Fas ligand (FasL) expression on Th1 cells, and the activation of natural killer (NK) cells. Based on these pleiotropic functions of IGIF, we named IGIF “IL-18” [2]. Although both IL-12 and IL-18 are major factors in IFN-γ production by Th1 cells, IL-12 is a differentiation factor that induces the development of Th1 cells—in contrast, IL-18 is a proinflammatory cytokine that facilitates IFN-γ production by Th1 cells particularly in conjunction with IL-12. Indeed, IL-12 and IL-18 from LPS-stimulated macrophages synergistically induced IFN-γ production from Th1 cells in *P. acnes*-primed and LPS-challenged mice.

Murine and human IL-18 proteins consist of 192 and 193 amino acids, respectively [1,3]. Based on the homology of its amino acid sequence compared with IL-1β, IL-18 is classified as a member of the IL-1 cytokine family. Human IL-18 and IL-1β share only 15% sequence homology although they contain a common β-pleated sheet structure. Furthermore, similar to IL-1β, IL-18 is produced as a biologically inactive precursor, pro-IL-18, which lacks a signal peptide and requires proteolytic processing to become active. The cleavage of pro-IL-18 or pro-IL-1β depends mainly on the action of the intracellular cysteine protease caspase-1 in the NLRP3 inflammasome [4,5].

The IL-18 receptor (IL-18R) consists of the inducible component IL-18Rα (IL-1 receptor-related protein [IL-1Rrp]) and the constitutively expressed component IL-18Rβ (IL-1R accessory protein-like [IL-1RAcPL]) [2]. Cytoplasmic domains of IL-18Rα and IL-18Rβ contain a common domain termed the Toll-like receptor (TLR)/IL-1R (TIR) domain, shared by other IL-1R family members and TLRs. Upon stimulation with IL-18, IL-18Rα forms a high-affinity heterodimeric complex with IL-18Rβ—which mediates intracellular signal transduction. Cytoplasmic TIR domains of the receptor complex interact with myeloid differentiation primary response 88 (MyD88), a signal adaptor containing a TIR domain [6], via TIR-TIR interactions. Then, MyD88-induced events result in the activation of nuclear factor (NF)-κB and mitogen-activated protein kinase (MAPK) via association with the signal adaptors IL-1R-associated kinase (IRAK) 1-4 and tumor necrosis factor (TNF) receptor-activated factor (TRAF) 6, respectively, which eventually leads to the appropriate gene expressions, such as *Ifng*, *Tnfa*, *Cd40l*, and *FasL*.

Although IL-18 was originally discovered as a factor that induces IFN-γ production from Th1 cells, it also acts on non-polarized T cells, NK cells, NKT cells, B cells, DC and macrophages to produce IFN-γ in the presence of IL-12. Moreover, IL-18 without IL-12 but with IL-2 induces Th2 cytokine production from CD4^+^ NKT cells, NK cells, and even established Th1 cells. Furthermore, IL-18 with IL-3 induces mast cells and basophils to produce IL-4 and IL-13. Therefore, IL-18 stimulates both innate immunity and acquired immunity [2,7].

The source of IL-18 was initially demonstrated to be from Kupffer cells, which constitutively express pro-IL-18. In addition, LPS binding to TLR4 induces the production of IL-18 via the activation of caspase-1. In contrast, upon stimulation with LPS, DC or macrophages increase their transcription of pro-IL-18 mRNA and subsequently their production of pro-IL-18, which is then processed by caspase-1 to be secreted as mature IL-18. In addition to these IL-18 producing cells, pro-IL-18 is produced by a wide variety of other cells, including keratinocytes, intestinal epithelial cells, and osteoblasts suggesting it has an important pathophysiological role in health and disease. Like other cytokines, IL-18 shows its pleiotropic action depending on its cytokine milieu (Figure 1).

## 2. Production of IL-18

Many cell types, both hematopoietic cells and non-hematopoietic cells, have the potential to produce IL-18. Originally, IL-18 production was recognized in Kupffer cells, liver-resident macrophages, even in the resting state without stimulation. However, subsequently, many investigators reported IL-18 production in non-hematopoietic cells, such as intestinal epithelial cells, keratinocytes, and endothelial cells, even in the steady state. In addition to its unique distribution and constitutive production in a wide variety of cell types and tissues, IL-18 is characterized by its unique process of cellular production. Usually, cytokines such as IFN-γ and IL-4 are secreted after the corresponding genes are expressed because their genes encompass a signal peptide that is necessary for their extracellular release through the endoplasmic reticulum to the Golgi. In contrast, the *IL18* gene, similar to other IL-1 family members, lacks a signal peptide. It was reported that IL-18 is stored in the cytosol of IL-18 producing cells [1,2,8]. Furthermore, similar to IL-1β but unlike IL-1α or IL-33, IL-18 is produced as a biologically inactive precursor [1,2,8]. To become active and be released, precursor IL-18 (pro-IL-18) needs post-translational processing [2,4,9]. Therefore, the extracellular release of biologically active IL-18 is regulated by multiple processes, such as regular transcriptional gene regulation, post-transcriptional gene regulation, and post-translational regulation.

### 2.1. IL18 Gene Expression

The *IL18* gene is located on chromosome 11 in humans and chromosome 9 in mice [2].

#### 2.1.1. Transcriptional Gene Regulation

##### 2.1.1.1. *IL18* Gene Promoter

The *IL18* gene contains 7 exons, where exons 1 and 2 are noncoding. An early study reported that promoter activity was detected upstream of exons 1 and 2 of the murine *Il18* gene [10]. Furthermore, the promoter upstream of exon 1 (5′-flanking region) contains an interferon consensus sequence binding protein (ICSBP)-binding site and activator protein-1 (AP-1)-binding site [11], while another promoter upstream of exon 2 (intron 1) encompasses a PU.1-binding site [11]. Similar to the genomic sequence of murine *Il18*, human *IL18* gene fragments were reported to contain a PU.1-binding site upstream of exon 2 and to have promoter activity [12].

A study on the detailed structure and sequence variations of the human *IL18* promoter revealed five single nucleotide polymorphisms (SNPs) at the 5′-end of the *IL18* gene: −656 G/T (rs1946519), −607 C/A (rs1946518), −137 G/C (rs187238), +113 T/G (rs360718), and +127 C/T (rs360717) [13]. The transcription activity of the *IL18* gene promoter fragment demonstrated that −656 G/T (rs1946519), −607 C/A (rs1946518), and −137 G/C (rs187238) are in the promoter region and that the other two SNPs are in the 5′-untranslated region (Table 1). A change from C to A at position −607 disrupted a cAMP-responsive element binding protein (CREB) binding site [13]. A change from C to G at position −137 altered the histone H4 gene-specific transcription factor-1 (H4TF-1) nuclear factor binding site [13] (Table 1). A new putative *IL18* gene variant was identified in systemic lupus erythematosus (SLE) patients [14]. These promoter variants were reported to reflect the protein levels of IL-18 produced by peripheral blood mononuclear cells (PBMCs) isolated from healthy individuals [15].

Intriguingly, many clinical study groups have extensively studied the association between these SNPs of *IL18* gene promoters and various diseases. Table 1 shows a summary of representative meta-analyses and/or systematic reviews of individual diseases. Therefore, *IL18* promoter variants are associated with diverse diseases such as chronic viral infection, chronic diseases, and cancer. Therefore, these promoter variants might influence pro-IL-18 production although they might not influence the release of biologically active IL-18. Therefore, how *IL18* promoter variants are associated with the risk of individual diseases remains to be elucidated. Cytoplasmic IL-18 might exert unknown actions on cellular properties that might influence disease risk.

##### 2.1.1.2. *IL18* Gene Repressor

B cell lymphoma 6 protein (Bcl6) was demonstrated to repress the *IL18* gene. Bcl6 was originally identified as a human proto-oncogene [16] and was recently demonstrated to be a master regulator of follicular helper CD4^+^ T cells [17]. A putative Bcl6-binding DNA located in the 5′-noncoding region at a site −2686 from exon 1 is a prerequisite for the Bcl6 repression of the expression of luciferase under control of the *IL18* promoter. In response to LPS, bone marrow-derived macrophages from *Bcl6*^−/−^ mice expressed higher levels of *Il18* than those from control mice [18].

#### 2.1.2. Post-Transcriptional Gene Regulation (miRNA)

MicroRNAs (miRNAs) are endogenous ~21 nucleotide-long noncoding RNAs that form a large family of post-transcriptional regulators of gene expression in metazoans and plants [19,20]. Humans have approximately 800 miRNAs, which participate in most cellular processes. However, changes in miRNA expression are involved in the pathogenesis of human disease. miRNAs interact with their mRNA targets by base pairing only using short sequences from these RNAs and mediate post-transcriptional gene regulation by translational repression or mRNA degradation. Multiple miRNAs in combination regulate their common target mRNA, whereas individual miRNAs regulate multiple different mRNAs. Therefore, individual miRNAs coordinate the expression of cellular proteins. The detailed mechanisms of post-translational regulation by miRNAs were extensively reviewed in recent articles [21,22,23].

Multiple miRNAs regulate *IL18* gene expression, directly or indirectly, and might be associated with disease and/or disease severity [24] as discussed in the following examples.

Bruton’s tyrosine kinase (Btk) is a cytoplasmic non-receptor tyrosine kinase, and its loss-of-function mutation was verified to be responsible for a humoral immunodeficiency named X-linked agammaglobulinemia with impaired B cell development [25]. Recently, Btk was shown to be involved in the stabilization of various cytokine mRNAs including *Tnfa* mRNA [25]. Upon stimulation with LPS, *BTK* was similarly induced in human macrophages and human fibroblast-like synovial cells. However, *IL18* was induced in the macrophages but not in the synovial cells. This was explained by the induction of miRNA-346 (miR-346) in synovial cells. Indeed, transfection with miR-346 antisense restored *IL18* induction in synovial cells, which was Btk dependent. Therefore, miR-346 negatively regulates *IL18* levels by reducing the induction of Btk expression, at least in synovial cells [26].

miRNA-197 (mi-R197) was shown to regulate *IL18* mRNA expression. Base-pair sequences were identified in miR-197 and *IL18* in THP-1 cells, a human macrophage cell line. THP-1 cells transfected with miR-197 showed lower *IL18* expression and secreted lower levels of IL-18 than control cells. Intriguingly, there was a negative correlation between disease stage of hepatitis B virus (HBV)-infected hepatitis (asymptomatic HBV carrier, chronic hepatitis, and acute on chronic liver failure) and miR-197 expression levels in peripheral blood mononuclear cells (PBMCs), and a positive correlation between disease severity and *IL18* expression levels in PBMCs. Therefore, miR-197 might be important for the reactivation of hepatitis by targeting *IL18* [27].

### 2.2. Post-Translational Regulation of IL-18 (Processing of pro-IL-18)

Pro-IL-18 requires cleavage by appropriate enzymes to become active. Caspase-1, originally designated as IL-1β-converting enzyme and now a member of the cysteine protease (caspase) family, is a major IL-18-processing enzyme. Processing of IL-18 with caspase-1 occurs in cytoplasmic inflammasomes. Caspase-8, a pro-apoptotic caspase, was recently shown to also be involved in the activation of IL-18. Various other proteases produced by killer lymphocytes, neutrophils, and mast cells have the capacity to appropriately cleave pro-IL-18 into biologically active IL-18.

#### 2.2.1. Caspases

Caspases (cysteine-aspartic proteases) are proteases responsible for important biological events including inflammation as well as programmed cell death such as apoptosis, necroptosis, and pyroptosis [28,29,30]. Caspases are produced as biologically inactive precursors (pro-), and require cleavage by other caspase members or itself to become an active caspase [29,30]. Caspase-1 is an essential enzyme for the conversion of pro-IL-1β and pro-IL-18 into mature IL-1β and IL-18, respectively [4,9].

##### 2.2.1.1. Caspase-1 (Inflammasome NLRP3, NLRC4, and AIM2)

Caspases are produced as biologically inactive precursors (pro-), and require cleavage by other caspase members or themselves to become active caspases [29]. The inflammasome is a large protein-complex generated in the cytoplasm after the activation of cells by appropriate stimuli [31]. Caspase-1 is activated within inflammasomes formed in the cytosol [31]. After stimulation with specific inflammasome activators, the corresponding monomeric pattern-recognition receptors (PRRs) gather together to assemble pro-caspase-1 with help from an adaptor protein, ASC (apoptosis-associated speck-like protein containing a C-terminal caspase-recruitment domain; PYCARD), which forms a wheel-shaped inflammasome [32,33]. Pro-caspase-1 in the inflammasome undergoes autolysis to become active caspase-1, which converts pro-IL-18 into mature IL-18. Then, active caspase-1 cleaves gasdermin D to liberate a pore-forming domain, N-terminal, leading to the liberation of mature IL-1β and IL-18 and eventually pyroptosis [34,35,36]. Many PRRs participate in the inflammasome. In this review article, we briefly introduce NAIP-NLRC4, AIM2, and NLRP3 inflammasomes. Excellent reviews provide detailed features of these inflammasomes [37,38,39,40,41].

The nucleotide-binding oligomerization domain, leucine rich repeat and pyrin domain containing 3 (NLRP3) inflammasome was originally demonstrated to be a cytoplasmic platform necessary for caspase-1 activation [42,43,44]. To activate the NLRP3 inflammasome in bone marrow-derived macrophages, two sequential signals are needed—the first signal, “priming”, is usually TLR-mediated, while the second signal, “activation”, is triggered by a wide variety of cellular responses. NLRP3 activators include highly diverse molecules such as inducers of K^+^ efflux, initiators of Ca^2+^ mobilization, microbial pore-forming toxins, endogenous or exogenous particulates, microcrystals, and endogenous metabolites [45,46,47]. However, how these biologically different signals activate the NLRP3 inflammasome is unclear. Recently, a common NLRP3 activation pathway was identified [48]. Zhong et al. demonstrated that macrophages began to synthesize mitochondrial DNA in response to TLR-mediated “priming”. Upon stimulation by subsequent NLRP3 activators including inducers of K^+^ efflux, Ca^2+^ influx, and particulates, these macrophages underwent mitochondrial insult and produced reactive oxygen species (ROS) in their mitochondria. During the activation process, newly synthesized mitochondrial DNA might be oxidized by ROS, and small fragmented oxidized mitochondrial DNA might translocate from the mitochondria to the cytosol through injured mitochondrial membranes. Indeed, transfection with oxidized mitochondrial DNA activated the NLRP3 inflammasome (Figure 2A).

The nucleotide-binding oligomerization domain (NOD)-like receptor (NLR) family of apoptosis inhibitory proteins (NAIPs), including human NAIP, and murine Naip1, Naip2, Naip5, and Naip6, are localized in the cytoplasm and sense bacterial components. Gram-negative bacteria exert pathological actions by injecting toxic molecules such as flagellin, a protein composed of flagella, via a syringe-like shaped type III secretion system consisting of a rod and needle [49]. Upon sensing these rods, needles, and flagellin, cytoplasmic NAIPs associate with NLR family CARD domain-containing protein 4 (NLRC4), initially designated as Ice protease-activating factor (IPAF), to generate the NAIP-NLRC4 inflammasome, which eventually leads to caspase-1-mediated IL-18 secretion (Figure 2B) [50,51,52]. Therefore, upon infection with Gram-negative bacteria such as *Salmonella typhimurium*, the resulting production of IL-18 might participate in host defense against certain bacterial infections [50,51].

Absence in melanoma 2 (AIM2) recognizes double-stranded (ds) DNA derived from microorganisms and host cells [53,54,55]. Infection with dsDNA viruses such as cytomegalovirus, induces free viral dsDNA in the cytosol of host cells. Certain intracellular bacterium, such as *Francisella tularensis*, exists in the phagosomes of macrophages and evade detection by entering the cytosol. These bacteria are regarded as healthy in the host cytosol. However, proteolytic enzymes activated in the cytosol damage the cytoplasmic bacterium cell wall, liberating free bacterial dsDNA, which activates the AIM2 inflammasome, leading to the release of biologically active IL-18. Host-derived dsDNA can activate caspase-1-mediated inflammation. Irradiation injured host cell DNA, and free host dsDNA activates the AIM2 inflammasome. In contrast to NLRC4, AIM2 directly senses dsDNA to activate the AIM2 inflammasome (Figure 2C).

In contrast to bone marrow-derived macrophages, Kupffer cells, resident macrophages in the liver, release IL-18 upon a single stimulation with LPS in the absence of priming [1]. To release mature IL-18, TLR4 [56], TRIF (TIR-domain-containing adapter-inducing interferon-β) [57], NLRP3 [57], ASC [58], and caspase-1 [4] are required [59]. We found that Kupffer cells produce ROS in response to LPS and that a ROS inhibitor completely inhibited LPS-induced-IL-18 release (our unpublished data). General irradiation to generate chimeric mice causes the deletion of hematopoietic cells including splenic macrophages but does not kill Kupffer cells [60,61]. Therefore, Kupffer cells might be different from bone marrow-derived macrophages in the context of their potential to generate ROS.

IL-18 maturation occurs by the activation of other types of inflammasomes, such as NLRP6 and NLRP9b inflammasomes, particularly in intestinal epithelial cells, which contribute to gut homeostasis and host defense, respectively (described below).

##### 2.2.1.2. Caspase-8 (upon Fas Ligation)

Fas is an extracellularly expressed cell death receptor, and its ligation with FasL leads to apoptotic cell death, a programmed cell death without cell membrane insult. We and others found that upon Fas engagement, neutrophils and macrophages release biologically active mature IL-1β and IL-18, respectively that is caspase-1 independent [62,63]. Because pan-caspase inhibitors prevent Fas-mediated IL-1β and IL-18 release, we assumed that caspases other than caspase-1 might process these precursor proteins. Recently, we and others revealed that caspase-8, an apoptosis-initiating protease [29], was involved in converting pro-IL-18 into mature IL-18 after the stimulation of macrophages by FasL [64,65,66]. We found that the Fas-mediated pathway for IL-18 release was important for host defense against bacterial infection. *Fas*^−/−^ mice were highly susceptible to *Listeria monocytogenes*, an intracellular Gram-positive bacterium that causes serious food-born infections in humans, with the impaired secretion of IL-18 and IL-1β [64]. Fas-mediated IL-18/IL-1β processing does not require NLRC4, NLRP3, or caspase-1, but does require ASC and caspase-8 [64,65]. Recently, we reported that IL-1β processed by Fas-mediated caspase-8 activation was profoundly involved in the development of Th17/Th1 cells, but not Th1 cells, during *L. monocytogenes* infection [67].

#### 2.2.2. Other Proteases Involved in the Production of Biologically Active IL-18

Several proteases other than caspases can cleave pro-IL-18 to generate biologically active pro-IL-18 fragments.

Proteinase 3 is a 29 kDa serine protease [68], which is mainly produced by neutrophils and stored in their azurophilic granules [68]. Human oral epithelial cells that constitutively produce pro-IL-18 release biologically active IL-18 after stimulation with neutrophilic proteinase 3, even in the presence of caspase-1 or pan-caspase inhibitors [69]. The proteinase-3 cleavage site of human pro-IL-18 was later identified to be at I^46^, while the N-terminal amino acid residue of mature human IL-18 processed by caspase-1 is Y^35^ [70].

Mast cells predominantly accumulate in the dermis of atopic-dermatitis model mice, skin-specific caspase-1 transgenic mice [71], as well as human atopic dermatitis patients [72]. The incubation of human pro-IL-18 with human mast cell chymase produced biologically active IL-18, but its N-terminal amino acid residue was I^56^, which is different from mature IL-18 cleaved by caspase-1 or by proteinase 3 [70].

Granzyme B is a serine protease with aspartic protease activity, and is produced mainly by NK cells and cytotoxic T lymphocytes (CTLs). Upon the recognition of target cells, cytotoxic lymphocytes inject granzyme B through a target cell membrane pore generated by perforin polymerization, which results in the apoptotic death of the target cells [73,74]. Granzyme B cleaves human pro-IL-18 at the same site as caspase-1, and the granzyme B-cleaved pro-IL-18 fragment has the capacity to produce IFN-γ [75]. Human CD8^+^ T cells isolated from PBMCs express granzyme B and kill human keratinocytes that constitutively produce pro-IL-18, accompanied by the release of mature IL-18 [76].

### 2.3. Regulation of Circulating IL-18 by IL-18-Binding Protein (IL-18BP)

IL-18-binding protein (IL-18BP) binds to IL-18 with a higher affinity than IL-18R and inhibits IL-18 signaling [77,78,79]. This suggests that IL-18BP levels determine the free IL-18 concentration. Adult-onset Still’s disease is a multi-systemic inflammatory disease characterized by upregulated IL-18 levels and downregulated IL-18BP in the circulation, mediated by an unknown mechanism [80]. Recently, miRNA profiles in the plasma of adult-onset Still’s disease patients were investigated, and the plasma levels of miRNA-134 (miR-134) were positively correlated with disease activity scores and decreased after effective treatment [81]. *IL18BP* was identified as its target mRNA [81]. Therefore, elevated miR-134 in adult-onset Still’s disease might induce the upregulation of IL-18 by inhibiting *IL18BP*, and therefore, miR-134 might be a potential biomarker for this disease.

## 3. IL-18 Signaling

### 3.1. IL-18 Receptor

IL-18 mediates its effects by signaling through its receptor, belonging to the IL-1R family, composed of an IL-18Rα chain (IL-18R1, IL-1Rrp) and IL-18Rβ chain (IL-18R accessory protein, IL-1RAcPL) [2]. Following the binding of IL-18 to IL-18Rα, IL-18Rβ then binds to form a trimer. The intracellular region contains a TIR domain in common with TLR, and MyD88 binds to TIR to transmit a signal into the cell. Although IL-18Rα alone can bind to IL-18, its affinity is low [82]. The IL-18Rβ chain is required for high-affinity binding and cell signaling [83]. When IL-18Rα binds to IL-18Rβ, it causes a conformational change, resulting in a high-affinity receptor [84,85].

IL-18R expression, induced by stimulation with IL-12 and IFN-α (human) in T cells and NK cells or by signal transduction and transcriptional regulation via STAT4 (Signal transducer and activator of transcription 4), is essential for potent IFN-γ production [86,87,88,89]. However, IL-18R is also expressed in basophils, mast cells, and CD4^+^ NKT cells in the steady state, all of which produce Th2 cytokines such as IL-4 and IL-13 in response to IL-18 stimulation [2,7]. IL-18R is also expressed by non-immune cells such as epithelial cells and nerve cells and is involved in cell survival and differentiation. The regulatory mechanism of IL-18R expression in these cells is poorly understood.

IL-18Rα is important for inflammatory cytokine production following IL-18 stimulation. However, although contradictory, the inflammatory response was exacerbated in IL-18Rα-deficient mice because IL-18Rα also binds to the inhibitory cytokine, IL-37 (IL-1F7). When IL-37 binds to IL-18Rα, it prevents IL-18Rα binding to IL-18Rβ, which blocks the transmission of activation signals into the cell. Instead, the IL-37/IL-18Rα complex binds to IL-1R8 (TIR8, SIGIRR), which promotes anti-inflammatory effects by activating STAT3 and transmitting an inhibitory signal [90,91,92,93].

### 3.2. IL-18 Signaling Cascade

Following the trimerization of IL-18/IL-18Rα/IL-18Rβ, MyD88 binds to the Toll-IL-1 receptor (TIR) domain of IL-18Rα and IL-18Rβ [6]. IRAK1 and IRAK4 are combined via the death domain of MyD88 to form a Myddosome [94,95,96]. Furthermore, following binding to TRAF6, inhibitor of κB (IκB) is degraded, and phosphorylated p65/p50 NF-κB translocates into the nucleus [97]. The MAPK cascade of Extracellular Signal-regulated Kinase (ERK), c-jun N-terminal kinase (JNK), and p38 is also activated (Figure 3) [98]. These signals induce IFN-γ production in Th1 cells and promote cell proliferation. IL-18 stimulation also induces the phosphorylation and activation of phosphatidylinositol-3 kinase (PI3K)/Akt/S6 and mammalian target of rapamycin (mTOR), as well as autophagosome formation and the expressions of Bcl-xL and Bcl2 [99,100]. Although PI3K suppresses inflammatory cytokine production by TLR signaling in myeloid cells [101], this signal enhances the proliferation and survival of NK cells. The PI3K/Akt pathway is also important for the survival of non-immune system cells, such as keratinocytes and neurons following IL-18 stimulation [102,103].

IL-18R signaling is similar to that of IL-1R/TLR. On the basis of IL-1R signaling, the signaling pathway after TRAF6 is thought to be transmitted as follows. K63-polyubiquitin chain (K63-pUb) is formed by the E3 ubiquitin ligase activity of TRAF6 to recruit TAB2/TAB3 [104] and NF-κB essential modulator (NEMO) [105]. K63-pUb also activates TAK1 and the phosphorylation of TAB1, IκB kinase (IKK) α, IKKβ, and MAPK kinase (MKK) 3/6 by TAK1 promotes the subsequent activation and nuclear translocation of MAPKs and NF-κB, inducing the expressions of various genes [105,106]. Regarding the activation of PI3K, the direct binding of PI3K and MyD88, or the B-cell adapter for PI3K (BCAP) might be involved [100].

Regarding differences in signaling between IL-1 and IL-18, TRIF-related adaptor molecule (TRAM, TICAM2) is thought to be involved in IL-18R signaling. In TLR4 signaling, MyD88 and TRIF function downstream. Then, TIRAP (Mal) connects TLR4 and MyD88, and TRAM links TLR4 and TRIF. However, TIRAP does not participate in IL-18 signaling [107]. Instead, TRAM directly binds to IL-18Rα, IL-18Rβ and MyD88 and transduces IL-18 signals into the cell [108].

### 3.3. IL-18 Binding Protein

IL-18 binding protein (IL-18BP) is an endogenous soluble factor that specifically inhibits the action of IL-18. IL-1 receptor family proteins are generally characterized by an extracellular portion consisting of three immunoglobulin (Ig)-like domains; however, IL-18BP contains a single Ig domain and is similar to TIR8 (SIGIRR, IL-1R8). In addition to mammals, various viruses possess highly homologous genes. IL-18BP contains a signal peptide and its protein is secreted without a transmembrane domain [77]. When the extracellular region of IL-33 receptor (ST2) is secreted, as soluble ST2, competes for the binding of IL-33 and ST2. However, IL-18BP does not correspond to the extracellular ligand binding domain of the IL-18 receptor, which is encoded by another gene, and therefore it is different from classical soluble receptors.

The affinity of IL-18BP for IL-18 is about 400 pM, similar to that for IL-18Rα/IL-18Rβ, and much higher than for IL-18Rα alone (10–50 nM). IL-18BP inhibits the binding of IL-18 to IL-18R and neutralizes IL-18 activity, thereby suppressing IFN-γ production and limiting Th1 cell responses [77,109]. For example, the administration of IL-18BP substantially reduced the pathology in mouse models of experimental arthritis, colitis, endotoxin shock, and type 1 diabetes, disease models in which IL-18 is important for the pathology [110,111,112,113,114]. Furthermore, transgenic mice overexpressing IL-18BP were protected from acute renal injury induced by ischemia-reperfusion [115]. In a colitis model, IL-18 acted directly on intestinal epithelial cells to enhance inflammation, thereby exacerbating colitis via excessive IL-18 stimulation in IL-18BP-deficient mice. In this case, the inhibition of goblet cell maturation was observed in the intestinal tract [116].

In healthy humans, IL-18BP is present in the serum at a 20-fold molar excess compared with IL-18 [114], suggesting IL-18BP provides a threshold at which IL-18 does not mediate its effects until the concentration of IL-18 reaches a point where it does not induce systemic excessive Th1-type immune responses against general infection with low pathogenicity. However, because IL-18BP-deficient mice were reported to have markedly decreased levels of IL-18 in their blood, IL-18BP might function as a carrier to maintain a constant blood concentration of IL-18 [117].

In autoimmune inflammatory diseases where IFN-γ is involved in the pathology, the concentration of free IL-18 was more important for determining the severity of disease compared with IL-18 bound to IL-18BP [114]. In Wegener’s granulomatosis and systemic lupus erythematosus, the serum levels of IL-18BP and IL-18 were high [114,118], but the level of IL-18BP was insufficient to neutralize IL-18 and the level of free IL-18 was higher than that of healthy individuals. Macrophage activation syndrome is another disease where the clinical and hematologic abnormalities correlate with elevated free IL-18 levels [119]. These studies have shown that IL-18BP therapy may be of clinical value in situations where excessive IL-18 stimulation appears to cause disease or to enhance its severity. Indeed, clinical trials to investigate treatment with IL-18BP for adult-onset Still’s Disease and NLRC4-associated macrophage activation syndrome, inflammatory diseases associated with high plasma IL-18 levels, are ongoing [120,121,122] (ClinicalTrials.gov Identifier: NCT 02398435, NCT 03113760).

IL-37 is another endogenous factor that suppresses the action of IL-18. IL-37 has high homology with IL-18, and IL-18 BP also binds to IL-37 [90]. Binding to IL-37 enhances the ability of IL-18BP to inhibit IFN-γ induction stimulated by IL-18. Of note, IL-37 has not been found in mice and human IL-37 binds to IL-18R with very low affinity. However, mice expressing human IL-37 showed marked anti-inflammatory effects especially related to LPS-induced cytokine production and dendritic cell maturation [93]. Furthermore, human IL-37 expressing mice were also resistant to dextran sulphate sodium (DSS)-induced colitis [123].

The amount of IL-18BP is highly regulated at the level of gene expression. Because IFN-γ increases the gene expression and protein synthesis of IL-18BP [124,125], IFN-γ invokes a negative feedback loop for IL-18-mediated inflammation. This concept is supported by clinical data where patients treated with IFN-α for hepatitis show elevated levels of IL-18BP [126,127]. Moreover, in patients with familial hemophagia, the ability of IFN-γ to induce IL-18BP was decreased and inflammation might have been exacerbated by insufficient negative feedback [128]. IL-27 functions has both inflammatory and anti-inflammatory effects, but similar to IFN-γ, it utilizes STAT1 for signal transduction and increases the production of IL-18BP as a negative feedback loop against inflammation in skin keratinocytes, fibroblasts, ovarian epithelial cancer cells, and leukocytes [129,130].

## 4. Physiological Roles of IL-18

### 4.1. Cytokine and Immune Cell Milieu Determines the Biological Action of IL-18

#### 4.1.1. IFN-γ Production

Naïve Th cells stimulated with Ag and IL-12 or IL-4 develop into IL-18R expressing Th1 or ST2 expressing Th2 cells, respectively. Therefore, the expressions of IL-18R and ST2 are convenient cell markers for Th1 cells and Th2 cells, respectively. IL-18, originally discovered as an IFN-γ-inducing factor, induces IFN-γ production from IL-18R expressing Th1 cells. IL-18 also induces IFN-γ production from NK cells and CD4^+^ NKT cells, which constitutively express IL-18R [2,7]. In general, IL-18-induced IFN-γ production by Th1 cells, NK cells and CD4^+^ NKT cells is strikingly enhanced by costimulation with IL-12. Moreover, IL-18 can synergize with IL-12 to induce IFN-γ production in DC, macrophages, and B cells [2,7]. It is well known that B cells develop into IgG1 and IgE-producing cells after stimulation with anti-CD40 antibodies and IL-4. We found that when B cells were stimulated with anti-CD40 antibodies and IL-4 in the presence of IL-12 and IL-18, these activated B cells produced IFN-γ that inhibited IL-4 dependent IgG1 and IgE production but enhanced IgG2a production [131]. T cells increase their expression of IL-18R in response to IL-12—similarly, IL-12-stimulated B cells also showed increased IL-18R expression and the production of IFN-γ in response to IL-18 and IL-12 [132].

How do IL-12 and IL-18 synergistically induce IFN-γ production? One major mechanism is the reciprocal induction of the other’s corresponding receptor expression [2]. Another mechanism is the synergistic induction of IFN-γ at the molecular level. The *Ifng* promoter contains a consensus sequence for NF-κB, AP-1 and the STAT4 binding site. IL-18 activates the IRAK/TRAF6 pathway resulting in the activation of NF-κB and AP-1, while IL-12 activates STAT4. In combination, these activate the *Ifng* promoter, resulting in the synergistic induction of IFN-γ production at the transcription level.

#### 4.1.2. Innate-Type Basophil and Mast Cell Activation by IL-3 and IL-18

Following the cross-linkage of FcεR1 by the Ag/IgE complex, mast cells and basophils produce Th2 cytokines, including IL-4 and IL-13. We investigated whether mast cells or basophils also had the potential to produce IFN-γ when stimulated with a combination of IL-12 and IL-18. Unexpectedly, basophils and mast cells derived from bone marrow cell cultures supplemented with IL-3 for 10 days expressed the IL-18Rα chain [133]. Furthermore, basophils produced large amounts of IL-4 and IL-13 concurrently in response to stimulation with IL-3 and IL-18. In contrast, mast cells mainly produced IL-13 in response to IL-3 and IL-18. The cross-linkage of FcεR1 with the Ag/IgE complex modestly increased IL-4 and IL-13 production by basophils and mast cells stimulated with IL-3 and IL-18. Furthermore, mast cells and basophils did not produce IFN-γ in response to any combination of IL-3, IL-18 and IL-12. Because IL-18 in combination with IL-3 stimulated basophils and mast cells to produce chemical mediators and Th2 cytokines, we speculated that IL-18 might induce allergic inflammation without help from the Ag/IgE complex. Therefore, we described the IL-3 plus IL-18-dependent activation of basophils and mast cells as “innate-type basophil and mast cell activation”.

#### 4.1.3. Innate-Type Allergic Inflammation by IL-18

We detected high levels of IgG1 and IgE in the sera of wild type mice and IFN-γ-deficient mice infected with *Strongyloides venezuelensis* at days 10 and 14 after infection. When a mixture of IL-12 and IL-18 was injected daily into these infected mice, IgG1 and IgE responses were only inhibited in wild type mice infected with *S*. *venezuelensis*. IL-12 and IL-18 stimulation induced T cells and B cells to produce IFN-γ, resulting in the inhibition of IL-4-dependent IgE production. To our surprise, the injection of IL-12 and IL-18 increased the serum levels of IgG1 and IgE in IFN-γ-deficient mice infected with *S*. *venezuelensis*. Most surprisingly, the daily administration of IL-18 or a combination of IL-2 and IL-18 induced a striking increase in the serum levels of IgE that was dependent on CD4^+^ T cells and IL-4/IL-4R/STAT6 [134]. Consistent with this result, transgenic mice overexpressing human caspase-1 in keratinocytes (KCasp1-Tg) produced IL-18 and IgE in the serum. Furthermore, they spontaneously developed atopic dermatitis (AD)-like skin lesions [71]. Therefore, we disrupted the gene encoding STAT6, which is required for the signal transduction of IL-4, or for IL-18 in KCasp1-Tg. Disruption of STAT6 resulted in no IgE production and did not affect the skin manifestations of KCasp1-Tg. In contrast, IL-18-deficient KCasp1-Tg mice had markedly diminished skin lesions. Therefore, the overproduction of IL-18 from keratinocytes induced AD-like skin lesions in the absence of IgE [71]. On the basis of these results, we described IL-18-induced allergic inflammation as “innate-type allergic inflammation” [71,135].

In collaboration with IL-2, but in the absence of IL-12, IL-18 stimulated NK cells, CD4^+^ NKT cells and splenic Th cells to produce IL-3, IL-9 and IL-13. Because IL-3 and IL-9 induce mucosal mastocytosis, we examined the capacity of mice injected with IL-2 and IL-18 to protect against infection with *S*. *venezuelensis*. C57BL/6 mice pretreated with IL-18 and IL-2 developed mucosal mastocytosis and had high levels of serum mMCP1 (mouse mast cell protease 1), an activation marker of mucosal mast cells. Furthermore, they could expel the intestinal nematode *S*. *venezuelensis*. Therefore, IL-18 is important for the expulsion of intestinal nematodes [136].

#### 4.1.4. Th1 Cells Produce IFN-γ and IL-13 in Response to IL-18

Although IL-18 and IL-12 increased IFN-γ production from Ag-stimulated Th1 cells, the injection of a mixture of IL-12 and IL-18 into IFN-γ-deficient mice infected with *S*. *venezuelensis* increased the serum levels of IgG1 and IgE. These results suggested that IL-18 and IL-2 might stimulate even Th1 cells to produce Th2 cytokines. Therefore, we examined the capacity of IL-18 and Ovalbumine (OVA) peptide to stimulate OVA-specific Th1 cells to produce Th2 cytokines. We found that Th1 cells stimulated with OVA and IL-18 produced both Th1 cytokines (IFN-γ) and Th2 cytokines (IL-9, IL-13). Furthermore, additional IL-2 stimulation enhanced the production of Th2 cytokines, suggesting Th1 cells can alter their gene expression pattern in response to OVA, IL-2 and IL-18. We designated these Th2 cytokine-producing Th1 cells as “super Th1 cells”. Intriguingly, after several rounds of stimulation with Ag, IL-2 and IL-18, IL-18Rα-expressing and T-bet-expressing Ag-specific Th1 cells expressed GATA3, and started to produce both IFN-γ and IL-13 [137]. We verified that GATA3 was essential for the induction of IL-13 in Th1 cells after their stimulation with Ag, IL-2 and IL-18.

### 4.2. IL-18 in Host Defense

Microorganisms can be classified as intracellular or extracellular. Generally, intracellular microorganisms are killed by cellular immunity, whereas extracellular microorganisms are eliminated by humoral immunity. IL-18 plays an important role in host defense against various infectious microorganisms because it strongly enhances the induction of IFN-γ, nitric oxide (NO), and ROS in phagocytes. In addition, IL-18 directly activates CD8^+^ T cells, which play a central role in viral clearance. Furthermore, because IL-18 activates Th2 cytokine production and granulocytes in the absence of IL-12, it also acts defensively in helminth infection.

#### 4.2.1. IL-18-Mediated Defense Against Extracellular Pathogens

##### 4.2.1.1. Bacteria Infection

For defense against extracellular bacteria, it is important to activate macrophages by IFN-γ produced from NK cells and Th1 cells, opsonization by antibodies and activation of complement, and phagocytosis and ROS production by neutrophils and macrophages.

Patients undergoing severe surgical stress, i.e., trauma, burns, or major surgery, suffer a loss of immunity and physical barriers such as the skin and intestines, thereby increasing the risk of infection such as sepsis [138]. If the host is infected with bacteria where the host defense system is weakened, it cannot suppress bacterial proliferation, which might cause fatal multiorgan damage. In a mouse study, Kinoshita and colleagues showed that multiple doses of IL-18 restored a state of reduced immune function after injury suggesting the medical application of IL-18 for infections. Multiple injections of IL-18 in burned mice significantly increased IFN-γ production from mononuclear cells and improved bacterial clearance and mouse viability after *Escherichia coli* infection [139,140]. IL-18 also enhanced host defense against *Pseudomonas aeruginosa* infection by enhancing IgM natural antibody production from liver B1 cells [141]. Such antibodies can opsonize bacteria and promote their uptake by phagocytes before bacterial antigen-specific antibodies are produced. IL-18 may also be useful for preventing serious complications of pneumococcal respiratory infections in immunocompromised patients [138].

Burns activate neutrophils and cause tissue damage, whereas phagocytosis and pathogen killing are decreased. IL-18 restored the burn-related decrease in activity of neutrophils and enhanced phagocytosis and ROS production to prevent infection by methicillin-resistant *Staphylococcus aureus* [141,142,143].

Overall, these data suggest that IL-18 treatment might be an alternative and useful treatment for infection by extracellular pathogens, even in immunocompromised individuals.

##### 4.2.1.2. Helminth Infection

Several types of intestinal parasitic nematodes are removed from the mammalian host Th2 responses [144]. *S. venezuelensis* is a rodent intestinal nematode that induces a Th2 type immune response in the host. It is removed mainly by the action of chondroitin sulphate and protease released from activated intestinal mucosal mast cells [145]. IL-18 induces mastocytosis by increasing the mast cell growth factor IL-3, thereby promoting the elimination of *S. venezuelensis* [136]. *Trichuris muris* is also expelled by mucosal mast cells activated by a Th2 response. However, in the case of *T. muris*, the immune response induced by infection varies by mouse strain. Mice that mount a Th1 type immune response following infection are susceptible, whereas mice that mount a Th2 type immune response are resistant. In C57BL/6 mice, IL-12 is produced by *T. muris* infection and IL-12 and IL-18 promote IFN-γ-producing Th1 cell differentiation. In IL-18-deficient mice that are resistant to chronic infection, Th2 cytokine-producing cells are increased in the lymph nodes in addition to Th1 cells. However, the administration of IL-18 inhibited Th2 cytokine production and prolonged infection [146,147]. A similar effect of IL-18 was also observed for *Trichinella spiralis* infection [148]. This indicates IL-18 inhibits the production of Th2 cytokines. Therefore, in a helminth infection in which a Th2 type immune response is important, IL-18 has a different role depending on the nematode type and may play an important role in host defense.

#### 4.2.2. IL-18-Mediated Defense Against Intracellular Pathogens

##### 4.2.2.1 Bacterial Infection

The ability of IFN-γ to increase the expression of inducible nitric oxide synthase (iNOS) is essential for killing intracellular organisms. Because NO production is critical for intracellular killing, IL-18, an IFN-γ inducer, plays an important role in controlling infection.

*Mycobacterium avium* is an intracellular microorganism that infects and proliferates in macrophages. In an *M. avium* infection mouse model, genetically susceptible BALB/c mice had decreased expressions of IL-12 and IL-18 and reduced IFN-γ/Th1 responses. In contrast, resistant DBA/2 mice showed the increased expressions of IL-12, IL-18 and IFN-γ [149] which correlated with the clearance of *M. avium*. Indeed, IL-18-deficient C57BL/6 mice had a severe infection and pathological changes that were significantly suppressed by treatment with exogenous IL-18 [150].

The role of IL-18 in host defense against *M. tuberculosis* infection has been demonstrated in studies using IL-18 knockout and IL-18 transgenic mice. IL-18-deficient mice are susceptible to *M. tuberculosis* infection, and IFN-γ levels in the serum, spleen, lung, and liver were lower than in wild type mice [151,152]. IFN-γ production by spleen cells stimulated with mycobacterial antigen was also impaired in IL-18 knockout mice. In contrast, IL-18 transgenic mice were more resistant to *M. tuberculosis* infection than control wild mice, and IFN-γ levels in the serum and production by mycobacterial antigen-stimulated splenocytes were increased. These data suggest the important contribution of IL-18 in the development of Th1 immunity [151]. In *M. tuberculosis* infected IL-18-deficient mice, IL-17 and MIP-1α were increased instead of decreasing blood IFN-γ and NO, and in the lung, the M2 macrophage markers Arginase-1 and Ym-1 were increase. Neutrophils are important for the control of *M. tuberculosis* in wild type mice, but *Il18*^−/−^ mice had increased accumulation of neutrophils as assessed histologically. However, because *Il18*^−/−^ mice cannot control *M. tuberculosis*, IL-18 might enhance the *M. tuberculosis* bactericidal activity of neutrophils [152].

In humans, the IL-18 promoter gene −137 G/C polymorphism is a risk factor for tuberculosis in the Chinese population, and PBMCs of the −137 GG type had lower IL-18 production compared with the GC and CC types [153]. Extensive case-control studies reported that a polymorphism of the IL-18Rα gene was associated with tuberculosis risk in people aged over 46 years old [154]. In addition, a SNP of the IL-18Rα promoter was associated with the methylation status of the gene and IL-18Rα expression, and the increased DNA methylation and decreased expression of IL-18Rα might be partially involved in the increased susceptibility to *M. tuberculosis*. In mice, even when IL-18Rα is deficient, there is no change in susceptibility to *M. tuberculosis*, indicating differences in susceptibility related to IL-18 between humans and mice. In humans, IL-37 uses IL-18Rα as a receptor to suppress immune responses. IL-37 levels were increased in the blood of *M. tuberculosis* patients, suggesting immunity is suppressed by a reduction in IFN-α production and the induction of M2 macrophage differentiation [155,156].

*L. monocytogenes* are Gram-positive intracellular bacteria that cause food listeriosis in humans. *L. monocytogenes* invade and multiply within the cytoplasm of various types of cells, including macrophages, epithelial cells, and hepatocytes. The role of IL-18 in resistance to *L. monocytogenes* infection is controversial. In a study using IL-18R antibodies, IL-18 potentiated the resistance of mice to *L. monocytogenes* infection [157]. Furthermore, IL-12- or IL-18-deficient mice lacked the ability to produce detectable levels of IFN-γ in the serum and showed significant and moderate *L. monocytogenes* sensitivity, respectively. *Ifng*^−/−^ mice and *Il12*^−/−^*Il18*^−/−^ mice had a higher sensitivity to *L. monocytogenes* infection than wild type mice [158]. Moreover, the administration of IL-18 decreased the number of bacterial cells in the liver and spleen of *L. monocytogenes*–infected mice [159]. However, Lockner et al. and Tsuchiya et al. reported that mice had increased resistance to *L. monocytogenes* infection because of the lack of IL-18 [160,161]. It is thought that IL-10 secreted from IL-18-stimulated NK cells was involved in the increase of *L. monocytogenes* susceptibility by IL-18 [162,163]. The influence of the presence or absence of IL-18 is thought to depend on the mouse strain and the infectious dose of *L. monocytogenes* used for infection.

*Listeria* infection induces IL-18 production by inflammasome-dependent and -independent mechanisms. Bacteria invade cells by listeriolysin O, which activates the caspase-1 inflammasome through NLRP3 and AIM2 and induces IL-18 production, which is also produced by Fas signaling independent of the caspase-1 pathway. Uchiyama et al. revealed the mechanism of the induction of IL-18/IL-1β production by *L. monocytogenes* infection via Fas. Upon *L. monocytogenes* infection, macrophages produce type I IFN, which induces the expression of the *Il18* gene. NK cells express FasL in response to *L. monocytogenes* infection that stimulates Fas expressed by infected macrophages. ASC, ROS and caspase-8 are activated in Fas-stimulated macrophages, which then secrete IL-18/IL-1β in their active forms [64]. Ly6C^+^ monocytes produce IL-18 in *L. monocytogenes*–infected mice, which is important for the expression of IFN-γ, but not granzyme B, by CTLs [164].

Recently, a new mechanism for IL-18 production by *Listeria* infection was reported. Macrophages infected with *L. monocytogenes* recognize lipoteichoic acid in bacterial cells by NLRP6, which recruits caspase-11 and caspase-1 via ASC to form an inflammasome that processes IL-1β and IL-18 to their active forms. At this time, NLRP6 and caspase-11 are induced by type 1 IFN similar to IL-18. Mice deficient for NLRP6, caspase-11, or IL-18 are more resistant to *L. monocytogenes* than wild type mice, but this effect is reversed by the administration of IL-18 [165].

Caspase-1 and IL-18 are important for host defense against *Shigella flexneri*, the causative agent of Bacillus dysentery. IL-18-deficient mice nasally infected with *S. flexneri* developed more severe inflammation in the lungs compared with wild type mice, because they could not eliminate the bacterial infection. These studies indicate that IL-18 is important for the induction of inflammation and the effective elimination of bacteria [166].

In a mouse model of *S. typhimurium* infection, the protective role of IL-18 was demonstrated by the treatment of infected mice with anti-IL-18 Ab or the administration of IL-18. In mice susceptible to lethal *S. typhimurium* infection, the administration of IL-18 increased survival and reduced bacterial tissue load [167]. A beneficial effect of exogenous IL-18 was not observed in mice deficient for IFN-γ. Caspase-1-deficient mice with *Salmonella* infection died more rapidly than wild type mice. Both caspase-1 substrates, IL-18 and IL-1β, are involved in the control of *S. typhimurium*. Infection experiments with *Il18*^−/−^ mice and the administration of recombinant IL-18 to *caspase-1*^−/−^ mice showed that IL-18 was not important for resistance to the intestinal phase of infection, but rather to the systemic infection [168].

Endogenous IL-18 is also involved in defense against infection by *Yersinia enterocolitica* [169,170]. In these models, IL-18 protected mice from lethality in an IFN-γ-dependent manner.

##### 4.2.2.2. Protozoan Infection

*Leishmania major* is an intracellular parasitic protozoan that mainly infects monocytic cells such as macrophages. The healing of lesions caused by *L. major* infection requires the induction and expansion of Th1 cells and NO production, an important effector molecule involved in the clearance of *L. major* via iNOS induction by IFN-γ [171]. IL-12 or IL-18 was administered to *L. major*-sensitive mice did not induce wound healing, but combined IL-12 and IL-18 completely protected footpad swelling in a NO-dependent manner. Treatment with anti-IL-18 neutralizing Ab reduced NO production by the downregulation of IFN-γ induced by *L. major* infection and markedly reduced the tolerance to *L. major* infection. In addition, IL-18-deficient mice on a *L. major*-resistant background developed large lesions during the early stage of *L. major* infection compared with wild type littermates but these were eventually resolved [172,173]. However, IL-18 enhanced Th2 cytokine production and exacerbated footpad swelling in *L. major* susceptible BALB/c mice [174]. Neutralization of IL-18 by IL-18BP suppressed Th2 cytokine production by IL-18 and alleviated footpad swelling.

*Trypanosoma cruzi* is an intracellular protozoan parasite that causes Chagas disease, mainly in Latin America. Following infection with *T. cruzi*, early inflammation of the invasion site occurs followed by hepatoma, splenomegaly after lymphadenitis, and in some cases, acute myocarditis develops and the patient dies. Chronic infections cause chronic myocarditis, gigantic esophagus, and a giant colon. IFN-γ produced after infection is important for determining resistance or susceptibility. Infection with *T. cruzi* also induces the production of IL-12 and IL-18 [175]. IL-12 is the major factor important for defense because absence of IL-18 did not affect the resistance against the protozoa [176,177].

However, in humans there is a correlation between the rs360719 gene polymorphism and *T. cruzi* infection (seropositive). rs360719 is located in the promoter region of the *IL18* gene and its polymorphism is thought to affect *IL18* gene expression by creating a binding site for the transcription factor OCT-1 [178]. Therefore, IL-18 might be involved in the resistance to *T. cruzi* infection.

Infection with *T. cruzi* can cause myocarditis. Mice infected with a Colombian strain of *T. cruzi* had increased blood IFN-γ and IL-12 levels, and in IL-18-deficient mice, the trafficking of inflammatory cells to the myocardium and the number of protozoa in the tissues were reduced [179]. In addition, the myocardium of patients with chronic Chagas disease had a high expression of IL-18 [180], and in Brazilian patients a correlation with the rs2043055 SNP was shown [181]. Therefore, although IL-18 is not involved in the resistance to *T. cruzi* infection, it might affect the pathology of myocarditis.

The protozoan, *Tritrichomonas musculis*, a commensal parabasalid newly identified in 2016, activates the host epithelial inflammasome and induces IL-18 release. This epithelial-derived IL-18 promotes Th1 and Th17 immunity in dendritic cells and provides protection against mucosal bacterial infection. Colony formation by *T. musculis* exacerbated the development of T cell driven colitis and sporadic colorectal tumor formation [182].

*Toxoplasma gondii* is an intracellular parasitic protozoan that spreads by ingestion and vertical infection. Oral or intraperitoneal infection of low doses (<20 cysts) of *T. gondii* induced IFN-γ production from Th1 cells and NK cells, which mediated the resistance of mice against infection [183,184]. However, oral infection with a higher inoculum dose (50–100 cysts) resulted in a deleterious Th1 cell response characterized by the development of severe small intestinal necrosis caused by the overproduction of proinflammatory mediators [185]. In immunodeficient mice, the injection of IL-18 decreased the number of parasites isolated from the tissue and correlated with an increase in NK cell activity [186]. IL-18 contributed to the development of immunopathological findings in the small intestine after the oral high-dose infection of *T. gondii* via the induction of IFN-γ production [187]. By this mechanism, IL-18 acts on IL-15-dependent NKp46^+^ NK1.1^+^ cells to induce CCL3 production, which is involved in the accumulation of CCR1 positive inflammatory monocytes [188]. IL-22 is also an important cytokine for ileitis induced by *T. gondii* infection. IL-22 acts on intestinal epithelial cells to induce IL-18 expression, which induces IL-22 production from innate lymphoid cells (ILCs). This positive feedback causes excessive IFN-γ production and marked neutrophil infiltration in the ileum [189].

Malaria is a major cause of morbidity and mortality, especially in sub-Saharan Africa. When the malaria parasite invades the host body via mosquito blood sucking, it first infects hepatocytes and then red blood cells. The early production of IFN-γ and TNF-α is crucial for protective immune reactions [190]. However, the excessive and prolonged production of proinflammatory cytokines is involved in the etiology of diseases associated with symptoms characteristic of severe malaria. The role of IL-18 in host defense against blood stage mouse malaria infection was reported using the non-lethal strain *Plasmodium yoelii* 265 and the lethal strain *P. berghei* ANKA. Infection by *P. yoelii* 265 or *P. berghei* ANKA increased the production of IL-18, IL-12p40 and IFN-γ. The administration of IL-18 to infected mice increased the infiltration of inflammatory cells, consisting of mononuclear cells and Kupffer cells, into the liver and decreased necrosis and pigment hemozoin deposition. Furthermore, serum IFN-γ levels were elevated, the onset of parasitemia was delayed, and the survival rate of infected mice was increased. The administration of anti-IL-18 neutralizing Ab to mice exacerbated infection with *P. berghei* ANKA, impaired host resistance, and shortened the mean survival time. In addition, *Il18*^−/−^ mice were more sensitive to *P. berghei* ANKA than wild type C57BL/6 mice. These data suggest that IL-18 has a protective role in host defense by enhancing IFN-γ production during blood stage infection by murine malaria [191,192]. IL-18 increased NK cell reactivity to IL-2 by increasing CD25 expression and inducing higher IFN-γ production [193].

Serum IL-18 levels are increased in human *P. falciparum* malaria patients [194]. When serum levels of IL-18 and IFN-γ were measured by dividing malaria patients into non-complicated, severe, and cerebral malaria, an increase in IL-18 levels was observed in all three groups. In cases with severe malaria, IL-18 tended to be high during the disease course. In addition, there was a significant correlation between the IL-18 level in severe malaria patients and the extent of parasitemia [195].

#### 4.2.3. IL-18-Mediated Viral Clearance

In addition to the potent induction of IFN-γ, IL-18 activates CD8^+^ T cells, which play a central role in viral clearance, suggesting a role for IL-18 in viral infection. Infection with influenza A virus induced the production of IFN-α/β, TNF-α, IL-1β and IL-18 in human peripheral macrophages [196]. Virus infected macrophage-derived IL-18 acts synergistically with IFN-α/β, inducing rapid IFN-γ production by T cells, resulting in the induction of Th1 immune responses. In a mouse model of *vaccinia virus* infection, the intraperitoneal injection of IL-18 significantly inhibited tail pock formation in BALB/c mice. *Ectromelia virus* belonging to the orthopoxvirus family is used as a mouse model of human smallpox. In IL-18-deficient mice, resistance to *E. virus* infection was attenuated, CTL numbers were decreased, and Th2 cytokine producing cells and Treg cells were increased. Therefore, IL-18 contributes to antiviral immune responses by promoting IFN-γ production and the induction of CTLs [197]. IL-18 induces HIV replication and in vitro experiments using the monocyte cell line U1 showed an increase in HIV-1 production after stimulation with the proinflammatory cytokines IL-1 and TNF-α or after exposure to IL-6 [198]. IL-18 increases HIV-1 production from U1 cells [199] and treatment with TNF binding protein or neutralizing anti-IL-6 mAb decreased IL-18 stimulated HIV-1 production, suggesting a role for IL-18 in HIV-1 pathogenesis. IL-18 levels are elevated in the sera of HIV-infected patients 3-to-6 years after infection, and decrease thereafter. The serum of HIV patients contains a large amount of TGF-β and the ability of blood IL-18 to induce IFN-γ is low. Because TGF-β inhibits IL-18 production from peripheral blood monocytes, IL-18 and TGF-β concentrations are inversely correlated [200]. In a herpes simplex virus (HSV-1) infection mouse model, the administration of IL-18 prior to infection markedly improved survival through T and B cell independent enhancement of IFN-γ induced NO. This effect was also observed in athymic nude mice and SCID mice, suggesting that IL-18 potentiates innate immunity [201]. Resistance to primary infection with herpes simplex virus 2 (HSV-2) was attenuated in IL-18-deficient mice or anti-IL-18 antibody treated mice, and the survival rate was reduced after infection [202]. Inflammatory monocytes recruited to infected sites produce type I IFN and IL-18 induced by HSV-2 infection. IL-18 is important for IFN-γ production from NK cells [203]. Rotavirus (RV) infects small intestinal epithelial cells, causing severe dehydrating diarrhea in children and moderate intestinal distress in adults. Bacterial flagellin treatment prevents RV infection in mice through adaptive immunity, interferons (IFN, type I and type II), NLRC4, and TLR5, a receptor for the flagellin receptor, all of which combine to cure chronic RV infection. This effect is suppressed in IL-18BP transgenic mice and *Il18*^−/−^ mice. The same protective effect was obtained by administering IL-18 and IL-22 instead of flagellin. The activation of TLR5 in dendritic cells by flagellin induced the production of IL-22 from ILCs via IL-23. Furthermore, flagellin also induced the production of IL-18, which was dependent on NLRC4 and independent of dendritic cells and ILCs. IL-22 induced protective gene expression in intestinal epithelial cells and IL-18 induced cell death by activating caspase-3 to eliminate RV-infected cells [204].

### 4.3. IL-18 in Metabolism

#### 4.3.1. IL-18 in Metabolic Homeostasis

Dietary nutrients, such as carbohydrates, proteins, and lipids, are digested and absorbed through the alimentary tract. Glucose and amino acids generated from carbohydrates and proteins, respectively are absorbed in the small intestine and transported into the liver via the portal vein. Absorbed fatty acids from dietary lipids are synthesized into triglycerides, which are transported into the liver. In the liver, these dietary units are appropriately catabolized and/or anabolized into minimal substances, which are consumed in individual cells to produce cellular components, participate in cellular activities and biological functions. The residual energy in each cell is collected and stored as triglycerides in adipose tissues. However, excess energy initiates systemic diseases, termed “metabolic syndromes”, such as type 2 diabetes mellitus, atherosclerosis, and acute myocardial infarction.

Initially, PRRs were thought to distinguish between non-self patterns (e.g., bacterial lipopolysaccharide, viral single-stranded RNA) and self-derived molecules. However, soon after their discovery, PRRs were demonstrated to sense various self-derived molecules, such as monosodium urate crystals involved in gout [44], calcium pyrophosphate dehydrate crystals involved in pseudogout [44], amyloid b in Alzheimer’s disease [205], and islet amyloid polypeptide in type 2 diabetes mellitus [206]. Now we know that a wide variety of self-derived molecules can activate the NLRP3 inflammasome (as described above). Therefore, the activated NLRP3 inflammasome might be involved in metabolic syndrome. Furthermore, NLRP3 inflammasome-mediated IL-1β has been identified as a key pathogenic cytokine that triggers and/or promotes various metabolic disorders. In contrast, IL-18 is beneficial for metabolic homeostasis. In this section, we will discuss the beneficial roles of IL-18 in metabolic homeostasis and introduce topics regarding the unique pathogenic action of IL-18 in the development of metabolic syndrome.

IL-1β and IL-18 belong to the IL-1 superfamily, and are activated and secreted via a cytoplasmic, multimeric complex termed the inflammasome, to promote proinflammatory cytokine effects. In contrast to IL-1β, IL-18 is a prerequisite for metabolic homeostasis. Several cellular and molecular mechanisms for the maintenance of metabolic homeostasis by IL-18 have been reported.

##### 4.3.1.1. IL-18 Regulation of Food Intake

IL-18-deficient mice spontaneously developed obesity and insulin resistance when fed a normal chow diet [207]. *Il18*^−/−^ mice developed obese diabetes mellitus with hyperglycemia, hyperinsulinemia, impaired glucose- and insulin-tolerance tests, and ectopic lipid deposition in the aorta wall. This was also observed for transgenic mice systemically overexpressing IL-18BP, a functional serum decoy receptor for IL-18 [207]. Intriguingly, food intake in *Il18*^−/−^ mice was significantly higher than that of control mice. Treatment with recombinant (r) IL-18 improved food intake in *Il18*^−/−^ mice. Zorrilla et al. reported that *Il18*^−/−^ mice fed a low-fat or high-fat diet (HFD) were hyperphagic as compared with control mice [208]. Furthermore, they found that the intracerebral ventricular administration of rIL-18 negatively regulated food intake in a dose dependent manner in HFD-fed wild type mice [209]. These reports suggest that IL-18 suppresses appetite and promotes energy expenditure even in people consuming an HFD.

##### 4.3.1.2. IL-18 Regulation of Energy Expenditure by Activating Thermogenic Adipose Tissues

*Il18*^−/−^ mice are characterized by impaired energy expenditure [208,209]. Recently, the mechanism for the regulation of energy expenditure by IL-18 was unveiled. Skeletal muscle cells are a prototype cell for energy expenditure [210]. Thermogenic adipose tissues, especially brown and beige adipocytes that are mitochondrion-rich cells, are secondary cell types that contribute to energy expenditure [211,212]. However, mitochondria in these cells cannot efficiently oxidize fatty acids, but dissipate energy as heat, thus contributing to energy expenditure [211,212]. Adipocytes in white adipose tissues, such as subcutaneous adipose tissue, have low numbers of mitochondria and are not thermogenic. Under certain conditions, however, white adipocytes can transdifferentiate into thermogenic beige cells [211]. Recently, it was reported that *Il18*^−/−^ mice had an impaired activation of brown adipocytes, and the beiging of white adipocytes [213]. This impairment might account for the impaired energy expenditure in *Il18*^−/−^ mice.

IL-33 is a pro-atopic innate immune cytokine and is a member of the IL-1 family. Of note, group 2 innate lymphoid cells (ILC2s) express the IL-33 receptor and IL-33 directly activated ILC2s to induce adipocyte beiging [214] via methionine enkephalin, a beiging-inducing molecule [215], as well as directly activating eosinophils to induce the beiging of adipocyte precursor cells [216]. Recently, certain populations of ILC2 were shown to express IL-18Rα but not the IL-33 receptor [217]. Skin-resident ILC2s dominantly express IL-18R, although some fat-resident ILC2s also express IL-18R [217]. IL-18 induction of adipocyte beiging might be explained by the activation of IL-18Rα-expressing ILC2s.

##### 4.3.1.3. IL-18 Activation of AMPK and Lipid Oxidation in Skeletal Muscle

Adenosine monophosphate-activated protein kinase (AMPK) is a sensor of intracellular energy status and maintains energy stores by catabolic and anabolic pathways [218]. Skeletal muscles experience drastic energy changes during exercise and AMPK participates in cellular energy control [218]. Cytokines such as IL-6, were reported to activate AMPK in the skeletal muscle. Lindegaard et al. demonstrated that IL-18 activated AMPK in skeletal muscle cells and skeletal muscle strips in vitro and ex vivo [219]. Consistently, *Il18ra*^−/−^ mice were reported to be highly susceptible to HFD, in terms of weight gain and insulin resistance [219]. Intriguingly, ectopic lipid deposition was induced in the skeletal muscle, but rarely in the liver, of HFD-fed *Il18ra*^−/−^ mice accompanied by impaired activation of the AMPK pathway. Therefore, IL-18 might maintain metabolic homeostasis in part by activating the AMPK signal pathway in skeletal muscle.

##### 4.3.1.4. IL-18 Processed by the NLRP1 Inflammasome Protects Against Metabolic Syndrome.

Recently, NLRP1 inflammasome-mediated IL-18 was reported to contribute to metabolic homeostasis in vivo [220]. It is well established that lethal toxins generated by *Bacillus anthracis*, a causative bacterium of anthrax, activate the NLRP1 inflammasome to release mature IL-1β and IL-18 [221]. Similar to *Il18*^−/−^ mice, *Nlrp1*^−/−^ mice spontaneously develop obesity (Figure 4A). Their metabolic alterations are characterized by ectopic lipid droplets in insulin-responsive organs, hepatic steatosis and adiposity in skeletal muscle, an increase in the size of adipose tissues, and systemically decreased lipolysis [220]. *Nlrp1*^−/−^ mice suffer from metabolic syndrome with impaired glucose tolerance [220]. IL-18 levels in visceral adipose tissues were significantly decreased in HFD-fed *Nlrp1*^−/−^ mice compared with control mice. Chimeric mice generated by bone marrow cell transfer revealed that the selective absence of NLRP1 in non-hematopoietic cells, but not hematopoietic cells, caused these alterations when fed an HFD (Figure 4B). In contrast, mice with a systemic gain-of-function mutation in *Nlrp1*, namely *Nlrp1^MUT^* mice, spontaneously developed lethal inflammation associated with the enhanced production of IL-1β. Furthermore, *Il1r*^−/−^*Nlrp1^MUT^* mice were resistant to the induction of inflammatory diseases [220]. Convincingly, plasma levels of IL-18 were still elevated in *Il1r*^−/−^*Nlrp1^MUT^* mice compared with *Il1r*^−/−^ mice. *Il1r*^−/−^*Nlrp1^MUT^* mice showed a loss of fat in various organs associated with enhanced lipolysis and low glucose tolerance. When fed an HFD, *Il1r*^−/−^*Nlrp1^MUT^* mice exhibited cachexia-like morbidity and mortality with hepatic focal necrosis and highly elevated plasma levels of IL-18 (10-60 ng/mL) (Figure 4C) [220]. The depletion of *Il18* rescued *Il1r*^−/−^*Nlrp1^MUT^* mice from the HFD-induced fatal changes and they exhibited a comparable phenotype to *Il1r*^−/−^ mice (Figure 4C) [220]. These observations clearly demonstrate the importance of IL-18 for metabolic homeostasis, in particular by controlling lipolysis and insulin sensitivity. Currently, endogenous agonists of the NLRP1 inflammasome remain unclear.

#### 4.3.2. Detrimental Role of the NLRP3 Inflammasome/IL-1β Axis in the Development of Metabolic Syndrome

It is well established that chronic inflammation, induced by proinflammatory cytokines including IL-1β processed by the activated NLRP3 inflammasome, is associated with obesity and resultant metabolic syndromes, such as type 2 diabetes mellitus, atherosclerosis, and cardiac vascular diseases, which are leading causes of death [222,223,224,225,226,227,228]. Many NLRP3 activators have been identified. First, oxidized low-density lipoprotein (LDL) was reported to activate macrophages through CD36, a scavenger receptor, to produce TNF-α and other proinflammatory cytokines, eventually leading to the development of atherosclerosis and cardiovascular diseases [229,230]. Under the conditions of oxidized LDL, cholesterol crystals that activate the NLRP3 inflammasome are generated in macrophages [231,232]. These events promote atherosclerosis. Second, saturated free fatty acids, such as palmitate, were identified as NLRP3 activators in addition to TLR4 agonists, which eventually cause type 2 diabetes [233,234]. Third, hyperglycemia activates the thioredoxin-interacting protein (TXNIP)-mediated NLRP3 inflammasome in adipose tissues and microvascular endothelial cells, which are relevant to type 2 diabetes and myocardial ischemic/reperfusion injury, respectively [235,236,237].

Patients with metabolic syndrome have elevated circulating levels of IL-18, a product of NLRP3 inflammasome activation [238,239]. Chronic inflammation in adipose tissues causes insulin resistance and type 2 diabetes mellitus. Notably, IL-1β, but not IL-18, participates in the development of metabolic syndromes by inhibiting adipocyte differentiation that is required for the maintenance of insulin sensitivity [240,241] and inducing inflammation [242]. IL-18 alone does not exert a proinflammatory effect when levels of co-activating cytokines, such as IL-12, are low [1,2,131,243,244]. Adipose tissue-associated macrophages of the M2 type produce anti-inflammatory and pro-fibrotic cytokines but low levels of proinflammatory cytokines [245,246]. This might explain the minor role of IL-18 in triggering obesity-associated metabolic diseases.

### 4.4. IL-18 in Intestinal Homeostasis

The gastrointestinal tract absorbs beneficial nutrients and paradoxically functions as a surface barrier to prevent harmful factors from entering the circulation. To transport dietary nutrients safely and efficiently, the gastrointestinal tract contains various control systems. Movement of the gastrointestinal tract is regulated by a unique nervous network termed the enteric nervous system as well as the central nervous system, under the control of sensing dietary contents, nutrients, and possible dietary toxins [247]. Once toxic foods enter the gastrointestinal tract, these nervous systems enhance gastrointestinal motility to rapidly exclude them via the production of diarrhea. In the gastrointestinal tract, especially the colon, various indigenous xenogeneic microorganisms, termed microbiomes, including viruses, bacteria, and fungi, reside symbiotically. Furthermore, the gastrointestinal tract transports food Ag and dietary nutrients into the tissues. Therefore, the gastrointestinal tract is always in contact with stimuli that activate innate and/or adaptive immunity. However, to accomplish their task of nutrient intake, the gastrointestinal tract negatively regulates immune responses directed to food Ags and microbiomes. Furthermore, symbiotic microbiome-derived metabolites, such as butyrate, are utilized to develop regulatory T cells that prevent pathological immune responses to food Ags and/or microbiomes in healthy individuals [248,249,250]. However, the gastrointestinal tract is a gate keeper, and should also detect and prevent the potential invasion of symbiotic microbes and food-borne pathogens [251,252]. Therefore, the gastrointestinal tract encompasses both a resistant barrier and a unique neural and immune system/network to prevent the entry on non-self factors.

Recently, IL-18 generated by intestinal epithelial cells was shown to be required for intestinal homeostasis [252,253]. In contrast, the uncontrolled release of IL-18 by immune cells infiltrating into lesions had an opposite detrimental effect in inflammatory bowel diseases (IBDs) [254,255]. Intriguingly, a recent article reviewed the importance of the NLRP3/IL-1β axis in IBDs [256]. Here, we introduce the beneficial roles of epithelial IL-18 in gut homeostasis.

#### 4.4.1. Intestinal Epithelial Cells Constitutively Produce Constituents of the Inflammasome for IL-18 Maturation

As described above, pro-IL-18 is produced and stored in a wide variety of cells. In contrast to pro-IL-18, pro-IL-1β is rarely expressed in the intestinal epithelium under normal conditions. Intestinal mucosal epithelial cells were reported to store pro-IL-18 in the cytoplasm [257,258,259]. These cells produce pro-caspase-1 and the caspase-1 activation adaptor molecule, ASC under a steady state [260,261,262]. Furthermore, several sensors or regulators of inflammasomes are also expressed constitutively, such as NLRP3 [263], NAIP-NLRC4 [264,265,266], and NLRP6 [267,268]. Therefore, intestinal epithelial cells have pre-formed components for inflammasomes. Of note, pro-IL-18, but not pro-IL-1β, is constitutively produced in tissue parenchymal cells in the steady state. Therefore, upon activation of the gut epithelial inflammasome, only mature IL-18 might be selectively activated and secreted. Therefore, the constitutive distribution of pro-IL-18 in specific cell types might determine the pro-homeostatic action of IL-18 on parenchymal cells.

#### 4.4.2. Importance of the Gut Microbiome for Homeostatic IL-18 Release

Gut microbiota are necessary for the appropriate development of innate and adaptive immunity. Indeed, intestinal T cells fail to develop into Th17 and Treg cells, under germ-free conditions [250,269,270,271,272]. However, supplementation with commensal bacteria or their products corrected this effect leading to their healthy development [250,269,270,271,272]. Recently, a microbiome-derived metabolite was demonstrated to be a prerequisite for the intestinal epithelial cell secretion of IL-18, which is necessary for intestinal homeostasis [273]. Mice fed a high-fiber diet were resistant to DSS-induced colitis. The microbiome catabolizes food into short chain fatty acids, such as acetate, in the mammalian gut. Mice deficient for G protein-coupled receptors that recognize short chain fatty acids, are highly susceptible to DSS challenge even when fed a high-fiber diet. Consistently, supplementation with acetate restored their resistance to DSS challenge. Intriguingly, the induction of IL-18 by acetate was dependent on NLRP3 but not NLRP6 [273]. These results suggest that metabolites generated by the microbiome participate in the constitutive secretion of IL-18 for gut homeostasis.

#### 4.4.3. Roles of NLRP6 Inflammasome-Mediated Epithelial IL-18 in Gut Homeostasis

*Nlrp6*, a member of the NLR family, is preferentially expressed in the kidney, liver, lung, and intestines, but rarely in the thymus or spleen [274]. In the colon, *Nlrp6* is expressed dominantly in epithelial cells, whereas *Asc* and *Caspase1* are similarly expressed in epithelial cells and CD45^+^ hematopoietic cells [267]. Similar to the NLRP3 inflammasome, the NLRP6 inflammasome is thought to consist of NLRP6, ASC, and pro-caspase-1 [261,275]. A recent report clearly demonstrated the requirement of caspase-11 for the optimal processing of caspase-1 by the NLRP6 inflammasome in macrophages upon transfection with lipoteichoic acids derived from a Gram-positive bacterium [276]. The loss of NLRP6 was reported to cause dysbiosis [261,267,275] and to impair the mucin layer in the colon of mice [277]. However, which stimuli initiate the sustaining activation of the NLRP6 inflammasome in intestinal epithelial cells remain to be elucidated. We need to identify whether lipoteichoic acid, presumably derived from the microbiome, contributes to the sustaining activation of the NLRP6 inflammasome.

##### 4.4.3.1. The NLRP6 Inflammasome is Indispensable for the Healthy Microbiota

Both *Nlrp6*^−/−^ and *Asc*^−/−^ mice are highly susceptible to DSS-induced colitis, strongly suggesting the importance of the NLRP6 inflammasome in this disorder [267]. Wild type mice co-housed with *Nlrp6*^−/−^ mice have a similar susceptibility to DSS challenge as observed for wild type mice co-housed with *Asc*^−/−^, *caspase1*^−/−^ or *Il18*^−/−^ mice [267], indicating colonic IL-18 is important for maintenance of the healthy microbiome (Figure 5). Indeed, the fecal bacterial phylogenetic architecture in these mice was clearly different from that in wild type mice [267]. Of note, wild type mice co-housed with *Il1r*^−/−^ mice show comparable clinical course after DSS challenge as wild type mice co-housed with wild type mice [267]. This clearly suggests that IL-1β is not generated by the NLRP6 inflammasome in colonic epithelial cells possibly because of the absence of pro-IL-1β. Alternatively, IL-1β generated through the NLRP6 inflammasome is not involved in the microbiome. Furthermore, wild type mice co-housed with *Aim2*^−/−^ mice or *Nlrc4*^−/−^ mice have an intact microbiome [267], suggesting that the AIM2 inflammasome or NLRC4 inflammasome are not activated in colon epithelial cells in the steady state. Therefore, the NLRP6 inflammasome is activated in the murine colon, leading to IL-18 release which maintains gut homeostasis by preventing dysbiosis.

Colon tissues from wild type mice under specific pathogen-free (SPF) conditions, but not under germ-free conditions, constitutively produce mature IL-18 [261]. Consistent with this, levels of IL-18 spontaneously produced by the colon of SPF mice are very low at birth, but are markedly increased at around 3 weeks, with continuous high levels of IL-18 thereafter [261]. Therefore, the microbiome might initiate IL-18 production (Figure 5). Upon stimulation with recombinant IL-18, colon tissues from germ-free mice produced antimicrobial peptides, which were beneficial for host defense on the barrier surfaces [278,279]. Colon tissues from wild type mice, but not *Asc*^−/−^, *Il18*^−/−^, *caspase1/11*^−/−^ or *Nlrp6*^−/−^ mice, constitutively produced antimicrobial peptides (Figure 5). Exogenous IL-18 restored the antimicrobial peptide production by these inflammasome-deficient mice concomitant with the restoration of dysbiosis [261]. Metagenomic sequencing of germ-free wild type mice co-housed with wild type mice or *Asc*^−/−^ mice revealed decreased taurine and increased histamine and spermidine in the metabolites of feces from the wild type mice co-housed with *Asc*^−/−^ mice compared with the wild type mice co-housed with wild type mice. The administration of spermidine or histamine in the drinking water impaired the spontaneous activation of caspase-1 and spontaneous production of IL-18 in the colon of wild type mice. Those observations suggest that NLRP6 inflammasome-mediated IL-18 sustains a healthy microbiome via the induction of antimicrobial peptides in the murine colon [275]. Endogenous ligands that activate the NLRP6 inflammasome for the continuous release of IL-18 remain to be elucidated.

##### 4.4.3.2. Involvement of the NLRP6 Inflammasome in the Formation of the Colonic Mucin Layer

The mucin layer is essential for intestinal host defense [280,281]. *Nlrp6*^−/−^ mice have colonic goblet cells numerically comparable to wild type mice. However, a detailed histological analysis showed that the width of the colonic inner mucus layer had almost disappeared in *Nlrp6*^−/−^ mice [277]. This was also observed in *Asc*^−/−^ mice and *caspase1/11*^−/−^ mice. In contrast, the inner mucin layer was intact in *Il1r*^−/−^ mice and *Il18*^−/−^ mice, indicating other biological events downstream of the NLRP6 inflammasome pathway might be involved in mucin formation (Figure 5). *Nlrp6*^−/−^, *Asc*^−/−^, and *caspase1/11*^−/−^ mice are highly susceptible to enteric infection with *Citrobacter rodentium*. Mice deficient in the NLRP6 inflammasome have impaired autophagosome formation [277] suggesting the NLRP6 inflammasome might contribute to healthy mucin layer formation by modulating the autophagosome pathway because the autophagy-mediated pathway contributes to other secretory pathways [282], and mice deficient in the molecule responsible for autophagosome generation have an abnormal inner mucin layer in the intestine [277]. The mechanism underlying the NLRP6 inflammasome induction of autophagy remains to be elucidated.

#### 4.4.4. Importance of Pyrin Inflammasome-Mediated Mucosal IL-18 for Tight Junction Formation

It is well established that mutations in *PYRIN*, alternatively named *MEFV*, are associated with a hereditary autoinflammatory disease termed Mediterranean fever as well as severe IBDs [283,284,285,286,287]. Pyrin is a member of the innate sensor family and forms part of the inflammasome after cells are stimulated by bacterial Rho GTPase [288,289]. *Pyrin*^−/−^ mice are susceptible to infection with pathogens that produce Rho GTPase as an exotoxin, including *Burkholderia cenocepacia* [288,289]. Intriguingly, ligands for TLRs, such as LPS, and TNF-α/TNFR1-mediated signaling induced *Pyrin* expression [290]. Recently, it was demonstrated that the pyrin inflammasome contributes to tight junction integrity to alleviate colitis and colitis-associated colon cancer in mice [290] (Figure 5). *Pyrin*^−/−^ mice are highly predisposed to colitis-associated colon cancer [290], which is induced by periodical treatment with AOM-DSS [291]. *Pyrin*^−/−^ mice had a larger burden of colon cancer with more severe colitis compared with wild type mice. Furthermore, colon tissues from *Pyrin*^−/−^ mice spontaneously produced higher amounts of proinflammatory cytokines/chemokines, but lower amounts of IL-18 despite comparable *Il18* expression, when compared with control mice. *Pyrin*^−/−^ mice exhibited a loss of epithelial barrier integrity with histologically and biologically impaired tight junctions, although the production of antimicrobial peptides and mucin were normal. Furthermore, the colons of DSS-treated *Pyrin*^−/−^ mice expressed higher levels of stem cell markers compared with DSS-treated control mice. Intriguingly, treatment with exogenous recombinant IL-18 restored epithelial permeability, colitis, and the tumor burden [291]. Therefore, in colon epithelial cells, the pyrin inflammasome might be constitutively activated to release IL-18, which sustains epithelial barrier integrity and prevents tumorigenesis. Intriguingly, *IL18* and *IL18RB* expressions were significantly lower in colorectal cancer patients than colon biopsies from healthy donors [291]. These observations suggest that the supplementation of IL-18 might be beneficial for certain types of colitis to protect against colorectal cancer.

#### 4.4.5. Newly Identified NLRP9b-Mediated IL-18 Release is Involved in Rotavirus Clearance

Very recently, Zhu et al. demonstrated that NLRP9b formed a new inflammasome comprised of ASC and pro-caspase-1, and that upon rotavirus infection it released IL-18, but not IL-1β, to induce pyroptotic cell death [41,292]. Pyroptotic cell death, but not IL-18, is important for rotavirus eradication. NLRP9b, similar to NLRP6, is constitutively expressed in intestinal epithelial cells. *Nlrp9b*^−/−^ mice, and mice selectively deficient in caspase-1 in epithelial cells, but not *Il18*^−/−^ mice, were highly susceptible to rotavirus infection, in terms of exaggerated diarrhea [292]. Furthermore, the NLRP9b inflammasome recognizes rotavirus RNA pathogen associated molecular patterns through DHX9, an RNA helicase [293]. Therefore, the gut mucosal epithelium is equipped with several inflammasomes to protect against various pathogens. Currently, whether NLRP9b is involved in the homeostatic production of IL-18 for intestinal homeostasis is unknown.

## 5. IL-18 in Disease

### 5.1. Endotoxin-Induced Systemic and Tissue Diseases

#### 5.1.1. Induction of Endotoxin Shock in *P. acnes*-Primed Mice.

Sepsis is still a common, life-threatening disorder, in which endotoxin is a key player. Paradoxically, patients with high serum levels of endotoxin do not necessarily develop lethal shock, whereas some patients die of septic shock even when their serum endotoxin levels are low. To understand this paradox, we measured serum IL-6 levels of patients, because LPS induces IL-6 production in vivo. The simultaneous measurement of serum levels of LPS and IL-6 indicated that there were at least two groups: the high IL-6 group was endotoxin shock susceptible and the low IL-6 group was endotoxin shock resistant [294]. These results suggested that limiting factors determine the sensitivity of patients to endotoxin shock.

Rodents are genetically resistant to LPS. Therefore, naïve BALB/c mice are resistant to challenge with high doses of LPS (100 μg/mouse). However, BALB/c mice primed with heat-killed *P. acnes*, a Gram-positive skin habituating bacterium, or BCG, become highly susceptible to the lethal shock-inducing effect of LPS. Furthermore, upon LPS (1 μg/mouse) challenge, they rapidly produced IL-1, TNF-α and IL-6 and died of endotoxin shock or, if they survived, they suffered from acute liver injury through apoptosis-mediated hepatocytotoxicity [294,295]. Moreover, *P. acnes*-primed mice became highly susceptible to the lethal shock-inducing effects of IL-1 and TNF-α, producing high levels of IL-6 and dying after challenge with IL-1 and TNF-α. Therefore, priming with *P. acnes* or BCG induced lethal endotoxin shock in mice highly susceptible to LPS by the enhanced production of IL-1, TNF-α, and IL-6.

After publishing these results [294], we observed that *P. acnes*-primed BALB/c *nu/nu* mice were resistant to LPS-induced lethal shock, but died of fulminant hepatitis [1]. However, *nu/nu* mice reconstituted with splenic T cells died of lethal shock before the development of fulminant hepatitis after sequential treatment with *P. acnes* and LPS [296]. Therefore, *P. acnes* pretreatment rendered mice highly susceptible to the lethal shock-inducing effect of LPS by the induction of Th1 cells. Indeed, IL-12p40-deficient mice or IFN-γ-deficient mice were highly resistant to *P. acnes*-primed and LPS-challenged endotoxin shock, revealing the importance of IFN-γ as a limiting factor to determine the sensitivity to LPS shock.

#### 5.1.2. LPS-Induced Liver Injury in *P. acnes*-Primed Mice

As noted above, *P. acnes*-primed and LPS-challenged *nu/nu* mice eventually died of fulminant hepatitis. However, the administration of anti-IL-18 Ab prevented LPS-induced liver injury in *P. acnes*-primed nu/nu mice [1]. We found that IL-18 induced FasL expression on Th1 cells, NK cells and unique liver T cells. Therefore, IL-18 is a key player in LPS-induced liver injury and induced fulminant hepatitis through Fas-mediated hepatocytotoxicity [295]. Indeed, *P. acnes*-primed IL-18-deficient mice were resistant to liver injury after LPS challenge. However, the administration of IL-18 induced liver injury in *P. acnes*-primed IL-18-deficient mice via the induction of FasL and TNF-α [63]. Therefore, we are very interested in how IL-18 is released after LPS challenge in *P. acnes*-primed mice [295].

Wild type mice primed with *P. acnes* developed dense granulomas in the liver, and developed acute liver injury when subsequently challenged with a sublethal dose of LPS [295]. These mice had elevated serum IL-18 levels after LPS challenge. Furthermore, *P. acnes*-primed IL-18-deficient mice exhibited granulomas in the liver comparable with *P. acnes*-primed WT mice, but were resistant to acute hepatitis induced by LPS. In contrast, MyD88-deficient mice, which lack signaling common to many TLRs as well as IL-18/IL-1β signaling, primed with *P. acnes* had low hepatic granuloma formation and undetectable levels of IL-18 after LPS challenge [57], although MyD88-deficient Kupffer cells secreted IL-18 in response to LPS in vitro [56,295]. Therefore, we examined the contribution of TRIF for *P. acnes*-induced hepatic granuloma formation and LPS-induced IL-18 secretion [57]. Unlike MyD88-deficient mice, *P. acnes*-primed TRIF-deficient mice normally develop hepatic dense granulomas, but do not release IL-18 or develop liver injury. Therefore, we concluded that *P. acnes* treatment induced hepatic granuloma formation that was dependent on MyD88. Subsequent LPS challenge activated caspase-1 via the NLRP3 inflammasome and induced IL-18 release, which was dependent on TRIF, eventually leading to liver injury [57].

### 5.2. IL-18 in Allergy

#### 5.2.1. Induction of IgE Production by IL-18

The daily administration of IL-18, especially with IL-2, markedly increased serum levels of IgE in naïve wild type mice [132]. An in vitro study revealed the increased expression of CD40L and production of IL-4 in CD4^+^NK1.1^+^ T cells stimulated with IL-2 and IL-18. These IL-18-stimulated NKT cells induced the development of naïve B cells into IgG1 and IgE-producing cells by the simultaneous stimulation of B cells with CD40L and IL-4 [134].

#### 5.2.2. Innate-Type Allergic Inflammation Induced by IL-18

We established transgenic mice overexpressing human caspase-1 in keratinocytes (KCasp1-Tg). These mice spontaneously produced IL-18 and IgE, and developed atopic dermatitis (AD)-like skin lesions. *Stat6-*deficient KCasp1 Tg mice did not produce IgE, but still developed similar skin lesions. Therefore, the overproduction of IL-18 from keratinocytes induces skin lesion even in the absence of IgE [71]. We described this inflammation as “innate-type allergic inflammation” [135].

#### 5.2.3. The Induction of IFN-γ and IL-13 Producing Super Th1 Cells by IL-2 and IL-18

Th1 cells produce both Th1 cytokines (IFN-γ) and Th2 cytokines (IL-9 and IL-13) in response to IL-18 plus IL-2. Furthermore, the intranasal administration of Ag, IL-2 and IL-18 to naïve mice bearing resting Th1 memory cells induced the development of airway inflammation and hyperresponsiveness [297]. We found that upon challenge with Ag, IL-2 and IL-18, resting memory Th1 cells produced both Th1 cytokines (IFN-γ) and Th2 cytokines (IL-9 and IL-13), which induced severe bronchial asthma. The administration of Ag and LPS also induced bronchial asthma by the induction of endogenous IL-18 from LPS-stimulated bronchial epithelial cells [298]. Therefore, Th1 cells, after stimulation with Ag and IL-18, become harmful cells that produce IFN-γ and IL-13, which induced difficult to control bronchial asthma [297,298]. We termed pathological Th1 cells as “super Th1 cells”, because they induced difficult to control asthma or AD-like skin lesions. This prominent feature of IL-18 might explain the mechanism for infection-associated allergic diseases.

#### 5.2.4. Bronchial Asthma Induced by the Intranasal Administration of IL-2 and IL-18

The nasal administration of IL-2 and IL-18 induced airway hyperresponsiveness, pulmonary eosinophilia, and goblet cell hyperplasia in wild type mice, but not in Rag2-deficient mice [299]. However, the nasal administration of IL-33 induced similar changes in wild type mice and Rag2-deficient mice [300]. Therefore, IL-2 plus IL-18 induced pulmonary changes in a T cell-dependent manner, while IL-33 treatment induced the same changes in a T cell-independent and innate cell-dependent manner.

### 5.3. IL-18 in Kidney Diseases

IL-18 is well documented as being involved in various types of kidney diseases. For example, mice deficient in *Il18* or those administered neutralizing anti-IL-18 Ab are resistant to acute kidney disease induced by ischemia/reperfusion [301] or by cisplatin treatment [302]. IL-18 blockade was also shown to protect against chronic kidney disease in mice induced by unilateral ureteric obstruction [303]. Recent excellent review articles have addressed this issue, in particular, focusing on its role in inflammasomes [304,305,306,307]. Here, we describe two topics of IL-18: its role in human IgA nephropathy and IL-18 as a clinical biomarker of acute kidney injury (AKI) that influences long-term outcomes of cardiac surgery.

#### 5.3.1. Association between Serum IL-18 Levels and Renal Prognosis in IgA Nephropathy

IgA nephropathy is a primary mesangial proliferative glomerulonephritis with the prevalent deposition of IgA in mesangial cells in the glomerulus. IgA nephropathy is regarded as a benign kidney disease. However, recent clinical studies revealed that IgA nephropathy had an extremely variable clinical course and that it led to end-stage renal disease with slow progression [308,309]. It was reported that serum IL-18 levels were a potent prognostic factor for IgA nephropathy [310]. Notably, serum concentrations of IL-18 in IgA nephropathy patients were significantly elevated compared with healthy controls [310]. Patients sensitive to corticosteroid therapy showed a significant reduction in serum levels of IL-18 after therapy, while patients resistant to therapy exhibited no reduction. Moreover, the renal survival of IgA nephropathy patients with higher than median serum IL-18 levels at baseline was approximately 20% at the end of the follow-up period (four years), and approximately 80% for total IgA nephropathy patients [310]. Furthermore, immunohistochemical analyses revealed that the intensity of IL-18 and NLRP3 proteins in renal biopsy samples from patients with IgA nephropathy correlated with the severity of proteinuria [311]. Therefore, serum IL-18 concentration might be a predictor for renal prognosis in this disease.

#### 5.3.2. Urinary IL-18 as A Biomarker of AKI after Cardiac Surgery

Because murine tubular epithelial cells secrete IL-18 and contain the components required for inflammasome activation [305,312,313], urine IL-18 levels might be elevated after acute tubular injury in human [314,315]. Urinary IL-18 is now recognized as a biomarker for AKI [316]. AKI often occurs in adults and children undergoing cardiac surgery and is a risk factor for morbidity and mortality [317,318]. Levels of serum creatinine, a biomarker for the diagnosis of AKI, increase late in the course of the disease, delaying timely treatment. Many studies have investigated new AKI biomarkers and several urinary proteins including IL-18 have been identified as early AKI biomarkers [319]. Recently, urinary biomarkers of AKI, particularly IL-18, were reported to be an additional prognostic factor for long-term postoperative mortality. Urinary IL-18 levels on post cardiac surgery days 1-3 were well correlated with the mortality rate at three-year follow-up [320]. AKI contributes to multiple organ failures [321,322], suggesting that long-term postoperative mortality might be directly evoked by AKI. However, a recent study revealed that postoperative AKI might be indicative of cardiac vascular stress, rather than an independent renal pathway for adverse cardiovascular death [323].

### 5.4. IL-18 in Metabolic Disorders

Early clinical studies revealed that IL-18 levels were elevated in the circulation and atherosclerotic plaques of patients with atherosclerosis [324,325]. In a prospective study of 1229 patients with coronary artery disease, at the 4-year follow-up, serum IL-18 levels were significantly higher in patients with fatal cardiovascular events than in those who did not die [326]. A community-based prospective cohort study showed that plasma IL-18 levels were a predictor of coronary evens in healthy European men [327]. Recently, a meta-analysis of the association of IL-18 with coronary heart disease identified circulating IL-18 as a possible risk factor of cardiovascular disease [328]. These reports suggest IL-18 is involved in metabolic syndrome. However, as described above (**4.3.2**), during metabolic disorders caused by excess energy, the NLRP3 inflammasome is likely to be activated by aberrant lipid metabolites and/or high glucose levels, which subsequently results in the secretion of IL-18 as well as IL-1β, which can induce inflammatory responses [242]. In contrast, IL-18 has a neutral or beneficial role in triggering obesity-associated metabolic diseases as described above (**4.3.2**). Therefore, an increase in circulating IL-18 concentrations might be an indicator of the activation levels of the NLRP3 inflammasome in the early phase of disease. IL-18, together with IL-12 and/or IL-15, exerts proinflammatory effects such as the activation of Th1 cells and the induction of IFN-γ by various immune cells including NK cells. During the progress of metabolic syndrome, abnormal metabolites, such as oxidized LDL and hyperglycemia activate the TLR4 and/or TLR2-mediated pathways [230,329], potentially leading to the production of IL-12 and/or IL-15. Under these conditions, IL-18 might activate NK cells and/or Th1 cells to produce large amounts of IFN-γ and/or TNF-α [1,2,132,243,244].

### 5.5. IL-18 in Cancer

As initially reported, IL-18 activates NK cells to produce IFN-γ and enhance cytotoxicity against tumor cells in synergy with IL-12 [1,243,244,330]. Because NK cells, and recently identified innate lymphoid cells, are well-established tumor-killing cells [331], many researchers have addressed whether IL-18 therapy rescues cancer expansion [332,333,334]. Here, we describe two recent theories on the beneficial roles of IL-18 in protecting against cancer. One is the establishment of cancer therapy by IL-18-activated human γδT cells. The other topic is the importance of fungi in microbiota for protection against colitis-associated colorectal cancer by inducing IL-18.

#### 5.5.1. IL-18 Robustly Expands Human γδT Cells

γδT cells have several innate cell-like properties [335,336]. For example, similar to αβT cells, γδT cells are activated upon T cell receptor (TCR) engagement. Whereas the TCR-mediated activation of αβT cells occurs in an MHC-restricted manner, the TCR engagement of γδT cells is independent of the MHC. To exert their biological function, naïve αβT cells require the appropriate differentiation into effector T cells, whereas γδT cells, including NK cells, can rapidly produce large amounts of cytokines and kill tumor cells. Indeed, γδT cells are well documented to exert tumoricidal activity [335,336,337]. However, low numbers of γδT ells are present in the peripheral blood of humans. Therefore, the bottleneck for the development of γδT cell-mediated cancer therapy has been the lack of an established method suitable for the efficient and safe expansion of γδT cells. Recently, Okamura’s group reported a protocol to obtain high numbers of γδT cells using IL-18 [338,339,340,341]. The incubation of human PBMCs including 1%–2% γδT cells with γδT cell Ag and IL-2 and IL-18, induced the proliferation of γδT cells, but not αβT cells, by approximately several thousand-fold in a 2-week culture [338,339,340,341] (Figure 6A). They used zoledronate as an activator of endogenous γδT cell Ag, which induces the accumulation of intermediate isopentenyl pyrophosphate, an endogenous γδT cell Ag, by blocking farnesyl pyrophosphate synthase in human monocytes [337,342,343]. The depletion of monocytes from PBMCs prevented the expansion of γδT cells [339]. Intriguingly, the IL-18-mediated expansion of human γδT cells requires CD56^+^CD11c^+^cells [338,339], initially termed NK-like dendritic cells (NKDCs) [344,345]. Indeed, CD56^int^CD11c^+^ cells in PBMCs co-cultured with monocytes in the presence of IL-12 and IL-18 robustly expanded and differentiated into CD56^bright^CD11c^+^ cells [338]. However, how CD56^bright^CD11c^+^ cells contribute to the expansion of γδT cells remains unknown. They also demonstrated that combination therapy with IL-18 and immune-checkpoint therapy with anti-PD-L1 and/or anti-CTLA4 mAb, synergistically prevented the mortality of mice harboring various tumor cell lines [346]. Combination therapy with IL-18 induced the expansion of precursor mature NK cells (counter cells of human CD56^+^CD11c^+^ cells) but did not affect regulatory T cells. The in vivo depletion of precursor mature NK cells or CD8^+^ T cells abrogated these therapeutic effects [346]. Therefore, IL-18 in combination with immune-checkpoint therapy might be a potential treatment for the early stages of cancer in humans.

#### 5.5.2. Mycobiome-Mediated IL-18 Protects Against Colitis-Associated Colorectal Cancer

A recent report confirmed the anti-cancer effect of IL-18 released from macrophages in response to commensal fungi on colitis-associated cancer [347]. The microbiome contains fungi as well as bacteria and other microorganisms [348]. C-type lectin receptors including Dectin-1, Dectin-2, Dectin-3, and Mincle are expressed on host cells and recognize β-glucan and α-mannans expressed by fungi [349,350,351,352]. Upon ligation with their ligands, C-type lectin receptors recruit Syk kinase, followed by NF-κB and MAPK signaling by assembling the CARD9/MALT/BCL10 complex [353]. Malik et al. reported that the recognition of commensal fungi by C-type lectin receptor induced Syk-dependent CARD9 inflammasome activation induced the release of mature IL-18 [347] (Figure 6B). *Card9*^−/−^ mice and mice selectively deficient for Syk in myeloid cells were predisposed to azoxymethane (AOM)/DSS-induced colitis-associated colorectal cancer, concomitant with reduced mature IL-18 in colon explants and an impaired accumulation of anti-tumorigenic T cells in the colon. Exogenous IL-18 prevented these mutant mice from colorectal cancer and restored the migration of anti-tumorigenic T cells. Of note, the administration of antifungal drugs rendered wild type mice highly susceptible to AOM/DSS-induced colitis-associated colorectal cancer, and supplementation with IL-18 rescued their predisposition to colorectal cancer [347]. These observations suggest the careful per os treatment of IBD patients with antifungal drugs might be of benefit. The depletion of mycobiota by antifungal drugs might initiate and/or promote colorectal cancer in IBD patients.

## 6. Similarities and Differences between IL-18 and IL-33

IL-33 is an IL-1 cytokine family member that uses ST2 and IL-1RacP (IL-1 receptor accessory protein) as a receptor to transduce signals via MyD88 [354], similar to IL-18. In contrast to IL-18R expressed on Th1 cells, which enhances IFN-γ production, ST2 is expressed on Th2 cells and enhances Th2 cytokine production. Furthermore, ST2 is expressed on various cells including ILC2s, mast cells, eosinophils, and basophils, and IL-33 stimulates these cells to induce Th2-type cytokines including IL-4, IL-5 and IL-13. By the action of these cytokines, IL-33 is involved in allergic inflammation and anthelminthic immunity by enhancing eosinophilic inflammation and mucus secretion [355].

Both IL-33 and IL-18 are released from epithelial cells or macrophages, but the individual production mechanisms are different. IL-33 exists as an active full-length form in the nucleus, which is released when cells undergo necrosis or are stressed [356]. In contrast, IL-18 stored in the cytoplasm as an inactive precursor and is released in response to stimulation by proteases such as caspase-1 [357]. When IL-18 is produced from macrophages by TLR stimulation, IL-12 may be also produced, to markedly induce the production of IFN-γ in vivo. Caspase-1 inactivates IL-33, and therefore, IL-33 is not produced when cells are activated to form an inflammasome by infection with bacteria harboring TLR ligands [358]. However, granule proteases in neutrophils and mast cells cleave IL-18 and IL-33 to enhance their activity [69,70,359,360]. IL-18 and IL-33 can stimulate basophils and mast cells to produce IL-4 and IL-13 and might be involved in Th2 type inflammation [300].

Mice overexpressing IL-18 or IL-33 in the skin developed spontaneous dermatitis, although their pathogenic phenotypes are different. Eosinophil infiltration was frequently observed in the lesion areas of skin-specific IL-33 transgenic mice [361], while neutrophil infiltration was mainly observed in skin-specific IL-18 transgenic mice [71]. The nasal administration of IL-2 and IL-18 induced airway hyperresponsiveness, pulmonary eosinophilia and goblet cell hyperplasia in wild type mice, but not in Rag2-deficient mice [299]. However, the administration of IL-33 induced these changes in wild type and Rag2-deficient mice [300]. IL-33 increased the number of ILC2s and promoted the production of IL-5 and IL-13 independent of T cells, thereby inducing lung eosinophilia and goblet cell hyperplasia. Therefore, IL-33 plays an important role in inducing Th2 cell-dependent and ILC2-dependent allergic diseases. ILC2s are activated by IL-33 during nematode infection. Infection of the lung by the intestinal nematode *S. venezuelensis* increases the number of IL-33-producing alveolar epithelium type II (ATII) cells in wild type and Rag2-deficient mice, which develop eosinophilic inflammation and goblet cell hyperplasia (Loeffler syndrome) [362]. Furthermore, ILC2s induced by *S. venezuelensis* became memory-like ILC2s, with a higher reactivity than ILC2s from naïve mice, and protected against new parasite infections by a non-specific mechanism [363]. As described above, lung ILC2s have strong reactivity to IL-33, but recently, differences in ILC2s isolated from specific organs were reported. ILC2s in the skin express IL-18Rα [364], and the effect of IL-18 on skin ILC2s is expected to be clarified in future studies.

## 7. IL-18 as A Therapeutic Target

Because of the strong proinflammatory activity of IL-18, many researchers are investigating IL-18 as a therapeutic target for the treatment of inflammatory diseases. To neutralize IL-18, IL-18 BP or anti-IL-18 Ab formulations were devised and clinical trials have been conducted to verify its safety and efficacy [365,366]. Clinical trials are underway to investigate the treatment of adult-onset Still’s disease and NLRC4-related macrophage activation syndrome (inflammatory diseases associated with high plasma IL-18 levels) using IL-18BP [120,121,122] (ClinicalTrials.gov Identifier: NCT 02398435, NCT 03113760).

In addition, the immunostimulatory effects of IL-18 have been investigated for treatments. The first attempts to administer IL-18 to cancer patients showed that its toxicity was generally mild-to-moderate [367]. For optimal cancer therapy, combination with other therapies is being considered. Anti-CD20 Ab is used to treat CD20 positive B cell lymphoma. Clinical studies using IL-18 with an anti-CD20 Ab are underway, and it was reported that the effect of anti-CD20 Ab was enhanced by the administration of IL-18 [368].

Although it is still at the stage of animal experiments, a new treatment method using IL-18 for cancer treatment has been studied. Recently, immune-checkpoint therapy by the neutralization of PD-1 or CTLA4 has dramatically improved cancer treatment. Combination therapy with IL-18 and an immune-checkpoint inhibitor synergistically reduced mortality in mice harboring various tumor cell lines. Therefore, IL-18 in combination with immune-checkpoint therapy might be a potential treatment for the early stages of cancer in humans [346]. Furthermore, chimeric antigen receptor (CAR) T cells artificially expressing a cancer antigen-specific TCR were effective treatments for B cell lymphoma and leukemia [369,370]. Studies on the effect on tumors of expressing IL-18 in CAR T cells in mice, demonstrated that IL-18 enhanced the antitumor effect [371,372]. These new therapy methods are expected to be applied to humans in the future and to save those suffering from cancer.

## Figures and Tables

**Figure 1 ijms-20-00649-f001:**
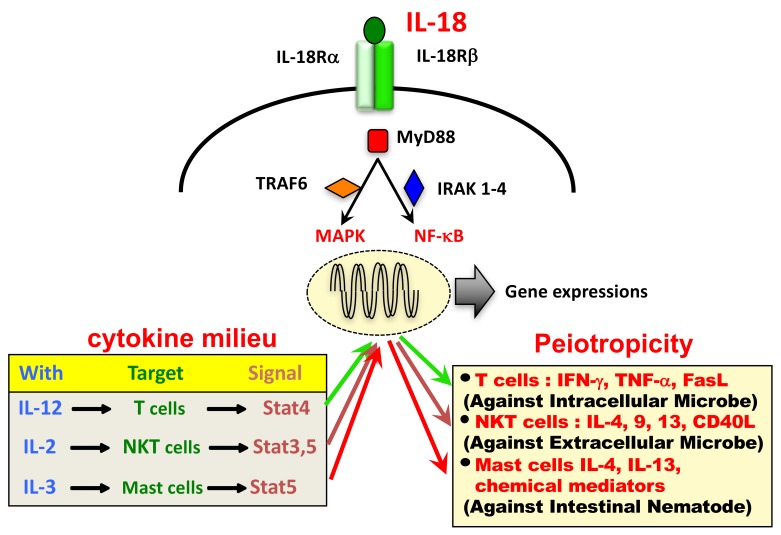
Pleiotropic action of IL-18 depends on its cytokine milieu.

**Figure 2 ijms-20-00649-f002:**
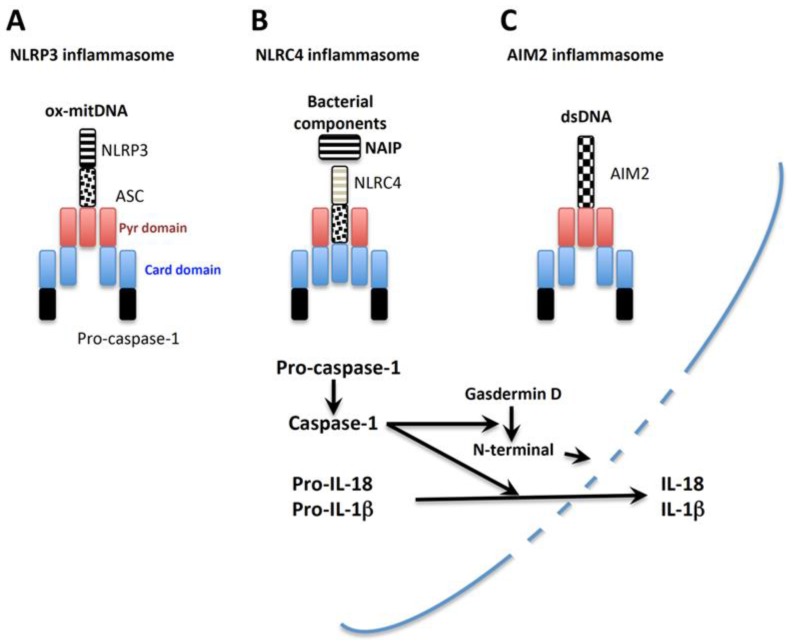
Representative inflammasomes. Caspase-1, an IL-1β/IL-18-converting enzyme, is produced as an enzymatically inactive precursor (pro). Upon the appropriate stimulation, many cell types including macrophages and epithelial cells, form cytoplasmic machinery termed the inflammasome for the activation of pro-caspase-1. Several cytoplasmic pattern-recognition receptors including NLR and AIM2-like receptors, serve as a scaffold molecule for individual inflammasome activation. (**A**) The NLRP3 inflammasome. After recognition of oxidized mitochondrial (ox-mit) DNA, NLRP3, a member of the NLR, assembles a caspase activating adaptor, ASC, and pro-caspase-1, which eventually leads to active caspase-1. (**B**) The NLRC4 inflammasome. After recognition of corresponding bacterial secretion system III, components from human NAIP or murine NAIP family members associate with NLRC4, which results in formation of the NLRC4 inflammasome. (**C**) The AIM2 inflammasome. After the recognition of double-stranded (ds) DNA, AIM2 similarly generates the AIM2 inflammasome. Active caspase-1 then cleaves pro-IL-18 and pro-IL-1β into IL-18 and IL-1β, respectively. Caspase-1 also cleaves gasdermin D into a pore-forming N-terminal protein. IL-18 and IL-1β as well as other cytoplasmic proteins including HMGB1, are extracellularly released through membrane pores generated by the N-terminal protein. Blue line indicated the cell membrane.

**Figure 3 ijms-20-00649-f003:**
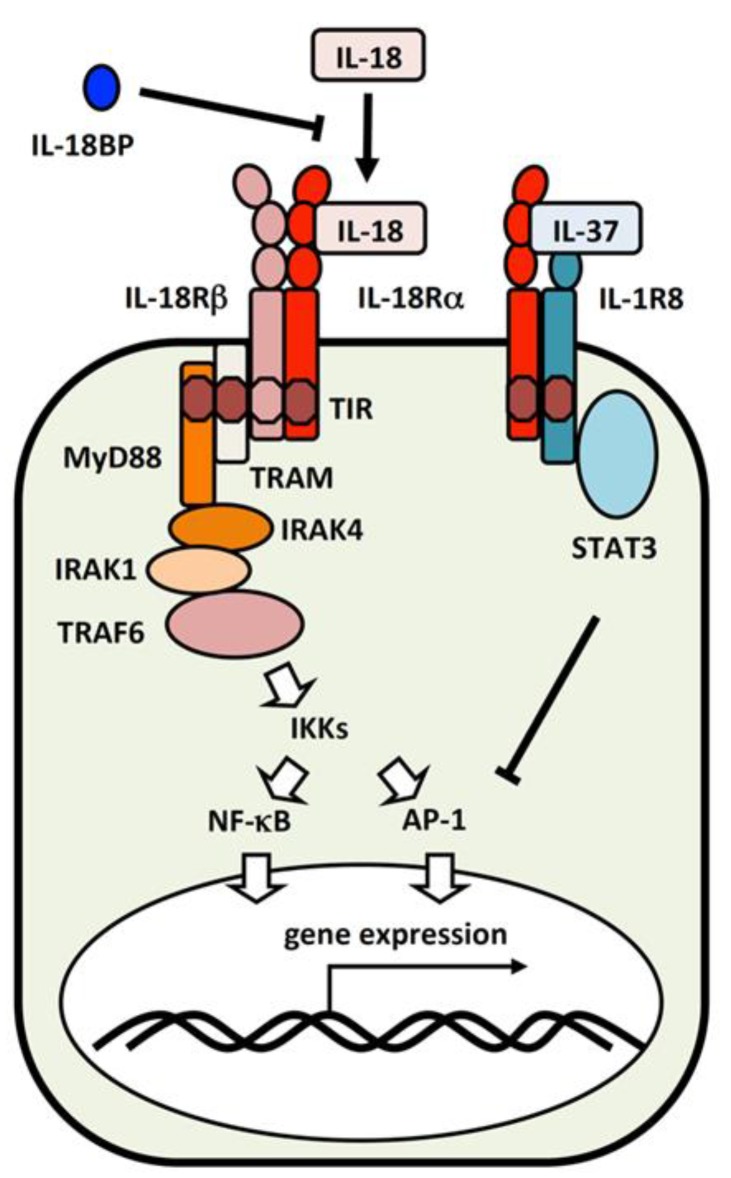
IL-18 signal transduction. When IL-18 binds to IL-18Rα (black arrow), IL-18Rβ binds to the complex. IL-18 signaling activates the transcription factors NF-κB and AP-1 via signal transduction molecules including MyD88, IRAKs, and TRAF6 (white arrows). IL-18BP competes IL-18 binding to IL-18Rα. When IL-18Rα binds to IL-37 it prevents binding to IL-18Rβ, which binds to IL-1R8 to send an inhibitory signal via STAT3.

**Figure 4 ijms-20-00649-f004:**
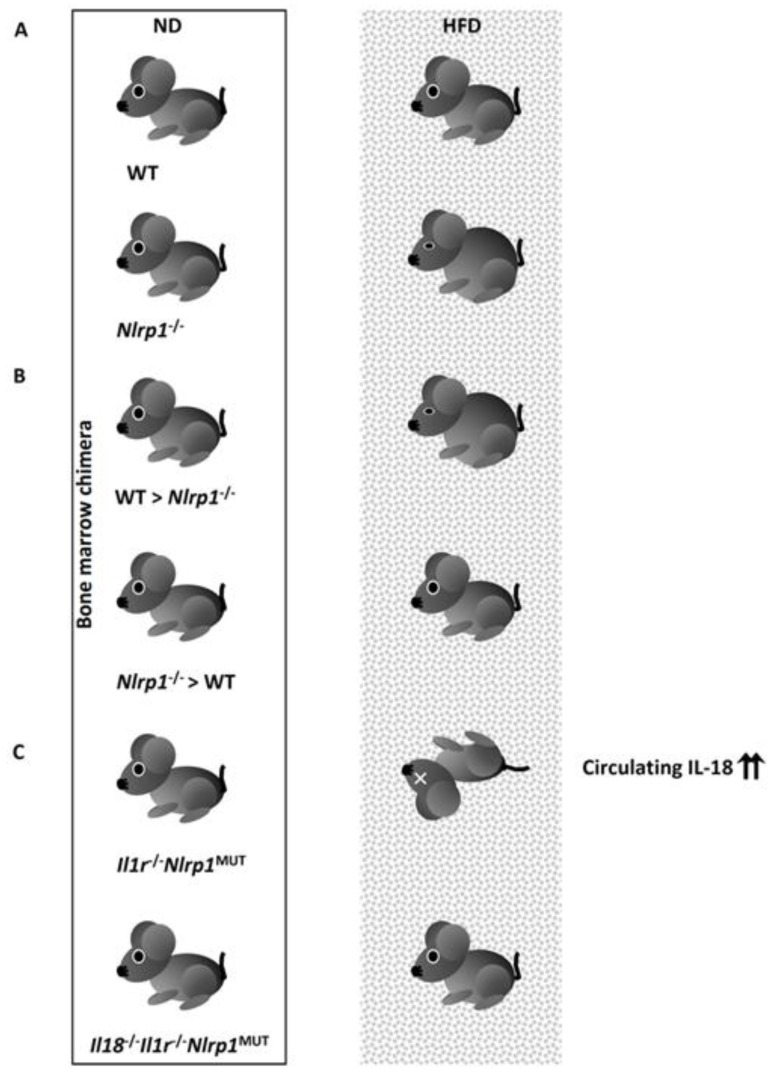
Involvement of the NLRP1 inflammasome in non-hematopoietic cells on metabolic homeostasis. (**A**) Requirement of NLRP1 for metabolic homeostasis: *Nlrp1*^−/−^ mice spontaneously develop obesity and related metabolic syndromes when fed a normal diet (ND) that does not induce obesity in wild type (WT) mice. *Nlrp1*^−/−^ mice fed a high-fat diet (HFD) had a more severe illness compared with WT mice. (**B**) IL-18 is released after activation of NLRP1 in non-hematopoietic cells. *Nlrp1*^−/−^ mice reconstituted with WT bone marrow cells (*Nlrp1*^−/−^ > WT) showed a phenotype similar to *Nlrp1*^−/−^ mice. In contrast, WT mice reconstituted with *Nlrp1*^−/−^ bone marrow cells (WT > *Nlrp1*^−/−^) exhibited a phenotype comparable with WT mice. (**C**) The protective role of IL-18 against cachexia. *Il1r*^−/−^*Nlrp1*^MUT^ mice harboring a mutant gene from human patients with familial Mediterranean fever but lacking the IL-1R gene have an almost intact phenotype when fed a ND. However, when fed an HFD, *Il1r*^−/−^*Nlrp1*^MUT^ mice suffered from fatal illness with a marked increase in circulating IL-18 levels. By contrast, *Il18*^−/−^*Il1r*^−/−^*Nlrp1*^MUT^ mice were resistant to HFD.

**Figure 5 ijms-20-00649-f005:**
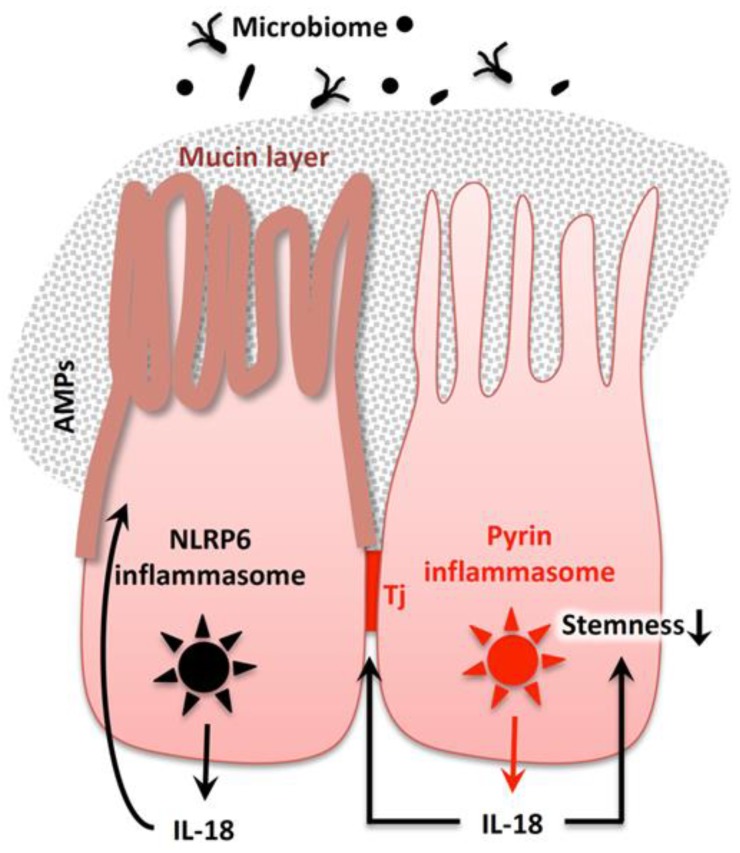
Importance of IL-18 for gut homeostasis. Intestinal epithelial cells utilize various inflammasomes for the constitutive production of IL-18. Under SPF conditions, colonic epithelial cells release IL-18 dependent on the NLRP6 inflammasome, which in turn activates epithelial cells to produce antimicrobial peptides (AMPs) to maintain microbiome homeostasis. *Nlrp6*^−/−^ mice spontaneously develop dysbiosis resulting in colitis. The NLRP6 inflammasome is involved in the generation of the homeostatic mucin layer by intestinal epithelial cells. IL-18 derived from pyrin inflammasome activation is involved in gut homeostasis through the generation of intact tight junction (Tj) formation. Furthermore, pyrin inflammation-mediated IL-18 protects against colitis-associated colorectal cancer by reducing the stemness of colon epithelial cells.

**Figure 6 ijms-20-00649-f006:**
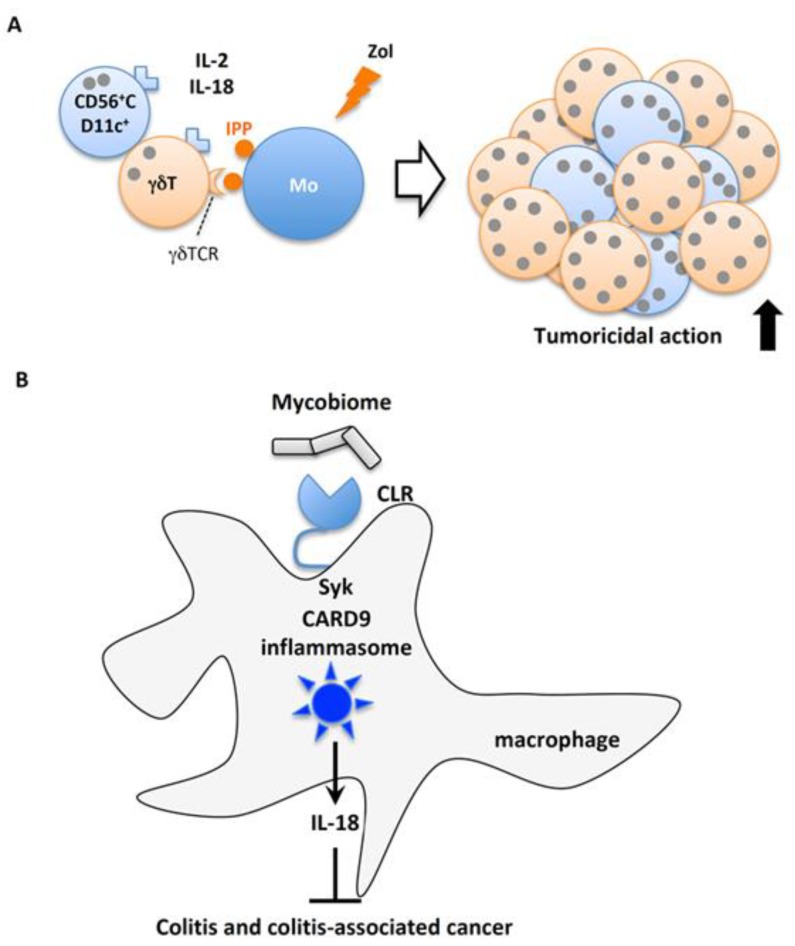
IL-18 protection against cancer. (**A**) Robust proliferation of tumoricidal human γδT cells. IL-18 in combination with IL-2 activates and induces the proliferation of human CD56^+^CD11c^+^ precursor NK cells, which in turn robustly proliferate and activate γδT cells stimulated with γδT cell antigen produced by zoledronate (Zol)-treated monocytes (Mo). (**B**) Involvement of CLRs-mediated CARD9 inflammasome activation in the induction of colon cancer. Fungi in the intestinal microbial flora activate macrophages through C-type lectin receptors (CLRs), which promote Syk to activate caspase-1 via the CARD9 inflammasome. The resultant IL-18 is required for protection against colitis and colitis-associated cancer.

**Table 1 ijms-20-00649-t001:** *IL-18* gene promoter polymorphisms (meta-analysis and/or systematic review).

Disease Association		SNP Polymorphism					Ref
		Promoters				5’-UTR	
		−1297 C/T	−656 G/T	−607 C/A	−137 G/C	+113 T/T	
		rs360719	rs1946519	rs1946518	rs187238	rs360718	
Chronic viral infection	HBV	ND	ND	+	+	ND	[373]
	HCV	ND	ND	−	+	ND	[374]
Periodontitis		ND	ND	+	+	ND	[375]
		ND	ND	−	−	ND	[376]
Autoimmune diseases	SLE	ND	ND	−	ND	ND	[377]
		+^a^	ND	+^a^	+^b^	ND	[378]
		ND	ND	+^c^	−	ND	[379]
		+	ND	+^a^	−	ND	[380]
	Behcet’s disease	ND	ND	+	ND	ND	[381]
	RA	ND	ND	−	ND	ND	[377]
		ND	ND	+^c^	−	ND	[379]
		ND	ND	+^b^	−	ND	[382]
	T1D	ND	ND	−	ND	ND	[377]
		ND	ND	+	−	ND	[383]
		ND	ND	+^b^	−	ND	[384]
	CD	ND	−	+	+	+	[385]
		ND	ND	−	ND	ND	[377]
	UC	ND	ND	−	ND	ND	[377]
Ischemic stroke		ND	ND	+	−	ND	[386]
Cancer	Total cancer	ND	ND	+	−	ND	[387]
		ND	ND	−	+^b^	ND	[388]
		ND	ND	+^b^	ND	ND	[389]
		ND	ND	ND	+^b^	ND	[390]
	Nasopharyngeal cancer	ND	ND	+	+	ND	[388]
		ND	ND	ND	+	ND	[387]
		ND	ND	ND	+	ND	[390]
		ND	ND	+	ND	ND	[389]
		ND	ND	+	ND	ND	[391]
	Esophageal cancer	ND	ND	+	ND	ND	[389]
	Gastric cancer	ND	ND	+	ND	ND	[388]
	Head and neck cancer	ND	ND	ND	+	ND	[392]
	Hepatocellular carcinoma	ND	ND	−	−	ND	[393]
Recurrent pregnancy loss		ND	ND	+	+	ND	[394]
		ND	ND	−	+	ND	[395]
Polycystic ovary syndrome		ND	ND	−	−	ND	[396]

+; significant association between the variant and disease. −; no significant association between the variant and disease. +^a^; association in European/Caucasian populations, but not in Asian populations. +^b^; association in Asian populations, but not in European/Caucasian populations. +^c^; association in Chinese populations, but not in Asian populations. ND; not determined. Abbreviations: HBV, hepatitis B virus; HCV, hepatitis C virus; SLE, systemic lupus erythematosus; RA, rheumatoid arthritis; T1D, type 1 diabetes mellitus; CD, Crohn’s disease; UC, ulcerative colitis.

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
