# Peer review of "Interleukin-18 in Health and Disease"

_ijms, 2019, doi:10.3390/ijms20030649_

Round 1
Reviewer 1 Report
The article entitled "Interleukin-18 in Health and Disease" was written in a clear way enriched with interesting drawings. In fact, most aspects of IL-18 in health and disease have been included in it. There is no subchapter about the role of IL-18 in chronic kidney disease, acute renal injury, IgA nephropathy and the role of this cytokine in cardiovascular prognosis in nephrology.
After considering this, I think the article will be suitable for publication in this Journal.
Author Response
Thank you for your important suggestion. We added a paragraph “5.3. IL-18 in kidney diseases” at pp28, after the paragraph “5.2.5. Bronchial asthma induced by the intranasal administration of IL-2 and IL-18”.
Reviewer 2 Report
The present manuscript is a review article about the cytokine Interleukin-18 in health and disease. The authors are experts for this cytokine, and the review is timely and will be of interest to a great readership. The review covers all aspects of IL-18 biology, is extremely comprehensive and well written. It is rare that I get a manuscript for reviewing that is of such high quality. Therefore, I have only some minor comments that the authors might want to take into consideration:
· The abstract is very detailed and not intuitive for people who are not experts on IL-18. My suggestion is to make the abstract easier to read and to remove all the specific details that are currently within the text.
· Line 94/96: “leader sequence” is an unusual term. “signal peptide” fits better.
· Line 149/150: The authors write “Bone marrow-derived Bcl6-/- macrophages upregulated Il18 compared with wild type cells” – I have difficulties to understand what this sentence means.
· Lines 263 ff.: The authors have written “(A)”, “(B)” and “(C)” – does this refer to Figure 2? This should be made clearer.
· After some sentences, the authors have a blank between the last word and the citation, in other the citation is directly after the last word. This should be written consistently according to the journals’ style.
· Although the review is already quite long, the authors should consider to add a paragraph to the end which deals with therapeutic opportunities. Would inhibition of IL-18 be beneficial, and if so, in which diseases? Could recombinant IL-18 be used in patients, and if so, in which? This would make a nice addition to the already very good review.
Author Response
· The abstract is very detailed and not intuitive for people who are not experts on IL-18. My suggestion is to make the abstract easier to read and to remove all the specific details that are currently within the text.
Thank you for your kind suggestion. We reexamined the summary carefully, and we removed some specific details from the abstract.
· Line 94/96: “leader sequence” is an unusual term. “signal peptide” fits better.
We corrected “leader sequence” to “signal peptide” as recommended.
· Line 149/150: The authors write “Bone marrow-derived Bcl6-/- macrophages upregulated Il18 compared with wild type cells” – I have difficulties to understand what this sentence means.
To clarify the information, we revised the sentence as follow:
“In response to LPS, bone marrow-derived macrophages from Bcl6-/- mice expressed higher levels of Il18 than those from control mice [18].”
· Lines 263 ff.: The authors have written “(A)”, “(B)” and “(C)” – does this refer to Figure 2? This should be made clearer.
These sentences are figure legend for Figure 2. We changed the font size of these sentences from “10” to “9” and moved them to the position immediately after the legend title of Figure 2 according to the journal template. We also corrected other figure legends for Figure 3-6.
· After some sentences, the authors have a blank between the last word and the citation, in other the citation is directly after the last word. This should be written consistently according to the journals’ style.
We added a space between the last word and the citation throughout the manuscript according to the journal’s style.
· Although the review is already quite long, the authors should consider to add a paragraph to the end which deals with therapeutic opportunities. Would inhibition of IL-18 be beneficial, and if so, in which diseases? Could recombinant IL-18 be used in patients, and if so, in which? This would make a nice addition to the already very good review.
Thank you for your important suggestion. We added a paragraph “7. IL-18 as a therapeutic target” at the end of the review.